# Touch in the Wild: Learning Fine-Grained Manipulation with a Portable Visuo-Tactile Gripper

**Xinyue Zhu**[1,*]    **Binghao Huang**[1,*]    **Yunzhu Li**[1]

[1]Columbia University

[*]Equal contribution

## Abstract

Handheld grippers are increasingly used to collect human demonstrations due to their ease of deployment and versatility. However, most existing designs lack tactile sensing, despite the critical role of tactile feedback in precise manipulation. We present a portable, lightweight gripper with integrated tactile sensors that enables synchronized collection of visual and tactile data in diverse, real-world, and in-the-wild settings. Building on this hardware, we propose a cross-modal representation learning framework that integrates visual and tactile signals while preserving their distinct characteristics. The learning procedure allows the emergence of interpretable representations that consistently focus on contacting regions relevant for physical interactions. When used for downstream manipulation tasks, these representations enable more efficient and effective policy learning, supporting precise robotic manipulation based on multimodal feedback. We validate our approach on fine-grained tasks such as test tube insertion and pipette-based fluid transfer, demonstrating improved accuracy and robustness under external disturbances. Our project page is available at https://binghao-huang.github.io/touch_in_the_wild/.

**Keywords:** Visuo-Tactile Sensing, Fine-Grained Manipulation, Scalable Data Collection

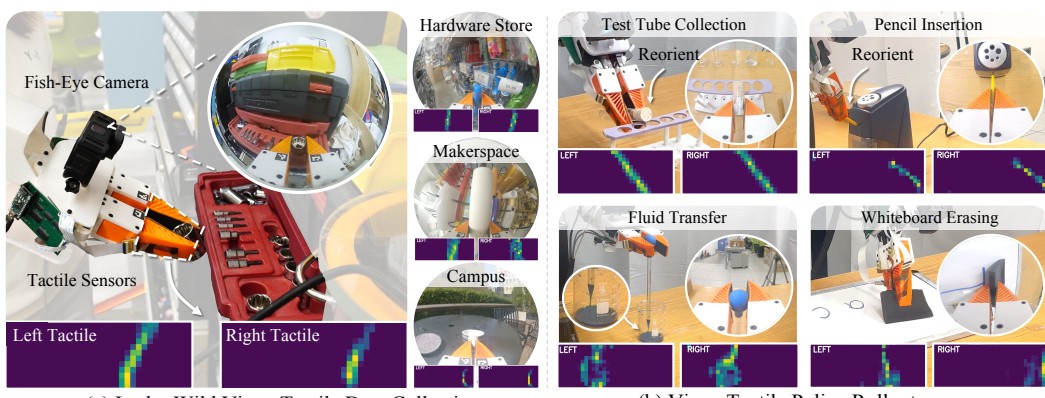

(a) In-the-Wild Visuo-Tactile Data Collection
(b) Visuo-Tactile Policy Rollout

Figure 1: (a) Our portable handheld gripper enables synchronized collection of visual and tactile data, supporting large-scale data collection in the wild. (b) We introduce a multimodal representation learning framework that fuses visual and tactile inputs to support fine-grained downstream manipulation tasks.

## 1 Introduction

Humans naturally rely on both vision and touch when interacting with the physical world. Whether inserting a key into a lock or adjusting a pipette during a lab experiment, tactile feedback plays a critical role in guiding precise motor actions, especially in situations where visual information may

39th Conference on Neural Information Processing Systems (NeurIPS 2025)

be unreliable due to occlusion, poor lighting, or dynamic backgrounds. While vision provides global, semantic context, touch offers local, high-resolution feedback about contact and force. The integration of these two modalities is fundamental to effective manipulation in everyday environments.

Recent advances in handheld grippers have made it easier to collect human demonstrations "in the wild." However, most existing systems focus exclusively on visual sensing, largely neglecting tactile feedback. This gap limits their usefulness for capturing the fine-grained, contact-rich strategies humans use in real-world tasks. Moreover, relying on vision alone makes these systems vulnerable to environmental variability, whereas tactile sensing offers a complementary and robust signal that is invariant to lighting conditions and camera viewpoint.

Two key challenges have prevented widespread visuo-tactile data collection in the wild: *(i) Portable tactile hardware.* Most existing tactile sensors are bulky, rigid, or not robust enough. For example, soft bubble sensors [1] are too large for handheld use, while other optical-based sensors [2] are often bulky or less robust for long-term in-the-wild use, making them less suitable for mobile or outdoor deployment. *(ii) Learning from multimodal data.* Tactile and visual signals differ significantly in scale and in the nature of collected information. Tactile inputs are local and physical, while vision encodes broader spatial context. Learning effective representations that integrate both modalities–particularly from large-scale, unstructured datasets–remains an open challenge.

To address these issues, we present a Portable Visuo-Tactile System for large-scale data collection and multimodal policy learning in real-world settings. Our contributions are threefold: *(1) A lightweight, handheld visuo-tactile gripper.* We integrate flexible piezoresistive tactile sensors into a soft, handheld gripper to enable portable, visuo-tactile, in-the-wild data collection. The system captures human manipulation demo trajectories across a wide range of environments, both indoors and outdoors. *(2) A multimodal representation and policy learning framework.* We introduce a masked autoencoding approach that uses cross-attention to jointly learn from visual and tactile inputs while preserving modality-specific characteristics. This design enables policies to interpret fine-grained tactile feedback more effectively, leading to improved sample efficiency and manipulation accuracy. *(3) An in-the-wild multimodal manipulation dataset.* We curated a diverse dataset of over 2.6 million visuo-tactile pairs, comprising more than 2,700 demonstrations across 43 manipulation tasks in 12 indoor and outdoor environments. This dataset supports effective visuo-tactile pretraining, and the resulting encoder significantly improves downstream policy learning. We are committed to open-sourcing the dataset.

We validate our system on fine-grained robotic manipulation tasks in the real world, such as test tube insertion and pipette-based fluid transfer, demonstrating successful policy transfer and robustness to environmental disturbances. Our results underscore the promise of portable visuo-tactile platforms in bridging the gap between human demonstrations and robot learning in complex, real-world settings.

## 2 Related Works

**Scalable multi-sensory data and learning.** Reinforcement and imitation learning have driven major advances in robotic manipulation [3–9], yet progress remains limited by the lack of tactile sensing and large-scale visuo-tactile datasets. Simulation can partially address data scarcity, but simulated tactile signals often diverge from real-world contact dynamics, reducing transfer effectiveness [5, 10–12]. This makes scalable real-world visuo-tactile data collection increasingly important. Acquiring multi-sensory data at scale, however, is difficult. Each additional modality—beyond RGB video—adds hardware complexity, synchronization overhead, and environmental constraints [13–18]. Consequently, most existing datasets are confined to structured indoor environments [19]. To address this, recent work has shifted focus toward large-scale visuo-tactile pretraining, which leverages existing data to learn shared representations that generalize across sensors and manipulation tasks [20, 21]. These approaches demonstrate that pretraining can reduce task-specific data demands and better align visual and tactile modalities, laying the foundation for scalable multimodal learning.

However, tactile sensing is most valuable in uncontrolled, in-the-wild settings, where vision may degrade due to poor lighting or background clutter [22, 23], while contact forces remain stable. Prior "in-the-wild" systems have demonstrated strong vision-only performance on simple tasks, but they

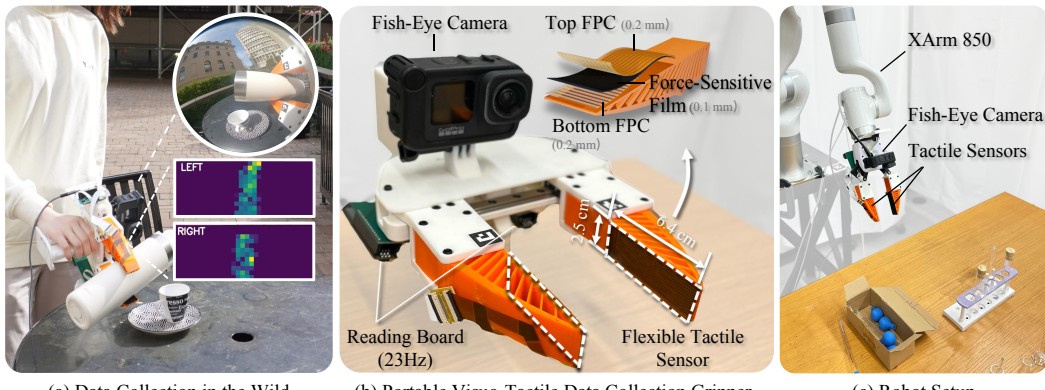

| (a) Data Collection in the Wild | (b) Portable Visuo-Tactile Data Collection Gripper | (c) Robot Setup |

Figure 2: (a) Multimodal data collection in the wild using our portable visuo-tactile system, with example tactile signals from both fingers. (b) Close-up of the handheld gripper, equipped with flexible tactile sensors and a fisheye camera for synchronized visuo-tactile capture. (c) Robotic setup for downstream tasks, featuring an XArm 850 with the same sensor configuration.

overlook the complementary benefits of touch [24, 25]. To bridge this gap, we propose a portable handheld visuo-tactile system that combines a fisheye RGB camera with a lightweight, flexible tactile array. This setup enables synchronized, large-scale collection of visual and tactile data across diverse, unstructured environments. Leveraging this system, we build a visuo-tactile dataset and train a masked reconstruction encoder that (i) accurately recovers tactile signals, and (ii) enhances downstream policy learning through visuo-tactile representations. In contrast to previous methods that directly merge vision and touch in a unified 3D space [19], our approach enables scalable cross-modal representation learning, fostering stronger correlations between visual and tactile modalities for fine-grained manipulation.

**Visuo-tactile manipulation.** Tactile feedback is essential for human manipulation, especially when vision is occluded or ambiguous [26, 27]. Likewise, tactile sensing enables robots to perform more adaptive and precise manipulation. Consequently, there is growing interest in combining vision and touch to enhance robotic manipulation [9, 19, 28–38]. Much of this prior work relies on optical tactile sensors, which image surface deformations to infer contact geometry and texture [15, 33, 39–43]. While these sensors provide rich signals, they are typically rigid, bulky, and less robust for long-term in-the-wild use, limiting their applicability in portable or unstructured settings.

In contrast, we use thin, flexible tactile sensors [19, 44, 45] that directly measure force distributions over the contact surface. Embedded in soft robotic fingers, these sensors provide consistent, object-agnostic representations that are easier to generalize and better suited for scalable, real-world learning. Although such sensors have been explored in lab setting [19, 46], large-scale, in-the-wild visuo-tactile datasets with synchronized RGB and tactile data capturing physical interactions have not previously been available. Our work fills this gap by collecting–and publicly releasing–a diverse, real-world visuo-tactile dataset. It spans a wide range of tasks and environments, laying the foundation for scalable multimodal learning and robust policy development. We acknowledge the concurrent project ViTaMIn [47], which introduces a portable visuo-tactile data collection system and a cross-modal learning framework for contact-rich manipulation. While our efforts share similar goals, our system design, representation learning strategy, and task focus differ in several key aspects. We appreciate their contribution and encourage readers to consult their work for complementary insights.

## 3 Visuo-Tactile Data Collection System

### 3.1 Scalable Flexible Tactile Sensors

As shown in Fig. 2 (a), we embed thin, matrix-based tactile pads into the soft, fin-shaped fingers of our handheld gripper. The sensor architecture builds on the triple-layer design from 3D-ViTac [19], adapted to fit the geometry of the adaptive fin-shaped gripper [24]. Each tactile pad consists of a piezoresistive sensing layer sandwiched between two flexible printed circuits (FPCs). To accommodate the elongated, flexible fins, we introduce two key modifications: *(1) Higher spatial*

*resolution.* Capturing contact patterns along the finger's length requires denser spatial sampling. The stainless-steel electrodes used in 3D-ViTac limit both resolution and signal stability. By replacing them with FPC electrodes, we achieve uniform trace pitch, improved robustness, and a per-pad resolution of $12 \times 32$ taxels, each measuring a $2 \times 2mm^2$ area. This allows us to capture fine-grained, dynamic contact signals. *(2) Rapid, scalable fabrication.* The use of FPCs enables tool-free assembly. Each pad can be fabricated in under five minutes and mounted on the gripper in an additional two, supporting scalable deployment for large-scale tactile data collection.

## 3.2 Portable Multi-Modal Sensing System

**In-the-wild large-scale data collection.** To enable real-world visuo-tactile data collection at scale, we design a compact and ergonomic handheld gripper that integrates both sensing modalities. Each tactile pad connects to a custom Arduino-based PCB, with two boards neatly housed beneath the gripper's palm (Fig. 2 (b)). The full handheld unit–including batteries–weighs approximately 962 g, making it comfortable for prolonged use.

At the firmware level, we optimize the serial protocol to stream each $12 \times 32$ pad at 23 Hz and synchronize with the visual information from the fish-eye camera. Tactile frames are timestamped directly on the microcontroller and transmitted over USB to a host device (e.g., a laptop or any portable Ubuntu system). The battery-powered, handbag-sized system is easily deployed in grocery stores, outdoor markets, and other unstructured, in-the-wild environments (Fig. 2 (a)), enabling high-throughput and scalable visuo-tactile data collection.

**Multi-modal data synchronization.** Precise alignment between vision and touch is essential for learning effective visuo-tactile representations. Although both the tactile system and the GoPro camera operate above 20 Hz (e.g., tactile sensors at 23 Hz, GoPro Hero 9 at 60 Hz), aligning their data streams poses challenges due to clock drift and limited timestamp precision on the camera.

We address this with a hardware-free synchronization strategy: *(i) Video stream.* Before each demonstration, a QR code displaying the current host time is shown to the camera, refreshed at 30 Hz. *(ii) Tactile stream.* Tactile data is published via ROS2 at 23 Hz, with each packet carrying a host-clock timestamp. *(iii) Post-processing.* During offline processing, we decode the QR code sequence from the video, recover exact host timestamps for each frame, and align them with the tactile data using the shared clock reference. This procedure yields tightly aligned visual and tactile recordings–without the need for external synchronization hardware–enabling accurate multimodal supervision for downstream learning.

## 4 Visuo-Tactile Representation and Policy Learning

To perform precise manipulation, robots must effectively integrate both visual and tactile signals. RGB images provide global, semantic context—such as object identity and workspace layout— while tactile signals offer local, contact-rich feedback that is often occluded from vision [48]. Because these modalities follow different statistical distributions, learning unified yet modality-specific representations for effective cross-modal reasoning remains challenging. We propose a two-stage learning framework (Fig. 3) that first learns a joint visuo-tactile representation via masked tactile reconstruction, and then integrates the learned representation into a diffusion policy [8] for downstream manipulation tasks.

## 4.1 Problem Formulation

Let $\mathcal{D}_{\text{pretrain}} = \{(I, T)\}$ be a large-scale dataset of synchronized RGB-tactile frame pairs, where $I \in \mathbb{R}^{3 \times 224 \times 224}$ is an RGB image captured from a wrist-mounted fisheye camera, and $T \in \mathbb{R}^{1 \times 24 \times 32}$ is a tactile image composed of vertically stacked fingertip sensor readings. The goal is to learn a multimodal encoder $E_\phi$, trained via self-supervised learning, that fuses these two modalities into a joint representation $z_{\text{fusion}} = E_\phi(I, T)$ that preserves modality-specific structure to support downstream manipulation tasks.

We divide the learning into two stages: *Stage 1:* Pretrain $E_\phi$ using a masked autoencoding objective that reconstructs full tactile images from partially observed tactile input and corresponding visual

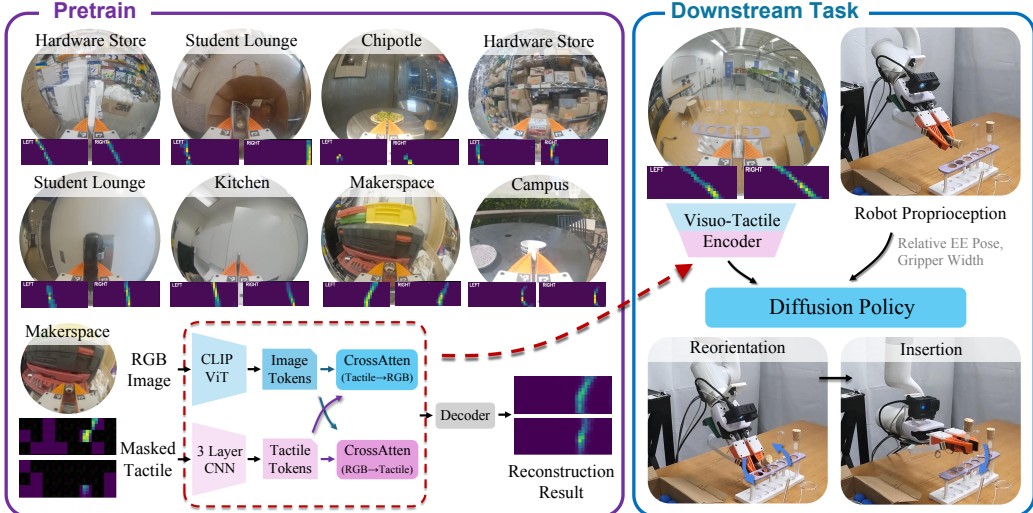

Figure 3: **Method Overview of Our Two-Stage Pipeline.** *Stage 1:* We pretrain a visuo-tactile encoder via cross-modal reconstruction using a large-scale dataset collected across diverse indoor and outdoor environments. *Stage 2:* The pretrained encoder is combined with robot proprioception to condition a diffusion policy for downstream tasks such as object reorientation and insertion.

frames. *Stage 2:* Use the pretrained encoder within a diffusion policy to learn manipulation behaviors from human demonstrations collected using our portable visuo-tactile gripper.

### 4.2 Stage 1: Visuo-Tactile Representation Learning

While prior work often relies on contrastive learning to align embeddings from different modalities [37, 47, 49], such objectives tend to suppress the fine-grained, geometry-sensitive signals captured by tactile sensors. Instead, we adopt a *masked autoencoding objective* [50], which reconstructs missing tactile regions conditioned on partially observed tactile input and visual context. This formulation encourages the encoder to retain tactile-specific information while leveraging vision for inference.

Formally, we jointly optimize the encoder $E_\phi$ and decoder $D_\psi$ via

$$(\phi^*, \psi^*) = \arg\min_{\phi, \psi} \; \mathbb{E}_{(I,T) \sim \mathcal{D}_{\text{pretrain}}} \left\| T - D_\psi \big( E_\phi(I, T) \big) \right\|_2^2, \tag{1}$$

where $E_\phi$ is our visuo-tactile encoder and $D_\psi$ is the tactile reconstruction decoder.

**Tactile encoder.** Each tactile reading consists of two fingertip arrays, each of shape $1 \times 12 \times 32$, stacked vertically to form a $1 \times 24 \times 32$ tactile image. We then apply a fixed colormap to convert this single-channel map into a 3-channel RGB tactile image. This image is divided into non-overlapping $4 \times 4$ patches, resulting in a $6 \times 8$ patch grid. During training, we randomly mask 60–80% of the patches in 95% of samples using a learnable token $T_{\text{mask}}$; the remaining 5% are shown in full. The masked tactile input is processed by a 3-layer CNN to produce a $d$-dimensional embedding $z_{\text{tac}}$, where $d = 768$.

**Vision encoder.** The RGB image $I$ is processed by a ViT-B/16 encoder initialized from CLIP [51]. We finetune all layers with a learning rate of $3 \times 10^{-5}$, and extract the final [CLS] token as the $d$-dimensional visual embedding $z_{\text{img}}$.

**Cross-modal fusion.** To integrate the tactile and visual features, we apply two rounds of multi-head cross-attention (MHAttn):

$$z'_{\text{tac}} = \text{MHAttn}(Q = z_{\text{tac}}, K = z_{\text{img}}, V = z_{\text{img}}) \xrightarrow{\text{LayerNorm}} z''_{\text{tac}}, \tag{2}$$

$$z'_{\text{img}} = \text{MHAttn}(Q = z_{\text{img}}, K = z''_{\text{tac}}, V = z''_{\text{tac}}) \xrightarrow{\text{LayerNorm}} z''_{\text{img}}. \tag{3}$$

We concatenate the updated embeddings to obtain the fused representation:

$$z_{\text{fusion}} = \big[ z''_{\text{tac}}; z''_{\text{img}} \big] \in \mathbb{R}^{2d}. \tag{4}$$

**Tactile reconstruction decoder.** The fused feature $z_{\text{fusion}}$ is passed through a two-layer MLP followed by a sigmoid activation to produce the reconstructed tactile image $\hat{T} \in \mathbb{R}^{1 \times 24 \times 32}$, i.e., $\hat{T} = D_\psi(z_{\text{fusion}})$. We use a full-image reconstruction loss $\mathcal{L}_{\text{stage1}}(\phi, \psi) = \|T - \hat{T}\|_2^2$, which encourages both local contact inference and global structural understanding.

**Stabilization via EMA.** We maintain a target encoder updated via exponential moving average (EMA) of the online weights with a decay factor of 0.9995. The tactile CNN and cross-attention layers are optimized with a learning rate of $1 \times 10^{-4}$, while the CLIP backbone is finetuned with $3 \times 10^{-5}$. The EMA encoder is used for checkpointing and attention visualization.

### 4.3 Stage 2: Policy Learning via Behavior Cloning

Once the visuo-tactile encoder $E_\phi$ is pretrained, it is integrated into a conditional diffusion policy for downstream manipulation tasks.

**Observation space.** At each timestep $t$, the robot first receives raw sensory inputs $(I_t, T_t, p_t)$, where $I_t$ and $T_t$ are the RGB and tactile images, and $p_t$ denotes the proprioceptive state (e.g., end-effector pose, gripper width). $I_t$ and $T_t$ are then passed through the pretrained encoder to produce the visuo-tactile embedding $z_t = E_\phi(I_t, T_t)$. The diffusion policy is then conditioned on $o_t = (z_t, p_t)$, i.e. the fused visuo-tactile embedding together with proprioception.

**Diffusion policy.** We adopt a conditional diffusion policy [8] that predicts noise instead of regressing actions directly. The model outputs $\hat{\epsilon}_t^k = \epsilon_\theta(a_t^k, o_t, k)$, where $a_t^k$ is the noisy action at step $k$, $o_t$ the conditioning observation, and $\hat{\epsilon}_t^k$ the predicted noise. During training, $k$ is uniformly sampled and Gaussian noise $\epsilon_t^k$ is added to the ground-truth action $a_t^0$. The loss is the MSE between added and predicted noise:

$$\mathcal{L}_{\text{stage2}} = \mathbb{E}_{t,k}\left[\|\epsilon_t^k - \hat{\epsilon}_t^k\|_2^2\right]. \tag{5}$$

At inference, actions are initialized as $a_t^K \sim \mathcal{N}(0, I)$ and iteratively denoised:

$$a_t^{k-1} = \alpha\left(a_t^k - \gamma\,\epsilon_\theta(a_t^k, o_t, k)\right) + \mathcal{N}(0, \sigma^2 I), \tag{6}$$

where $\alpha, \gamma, \sigma$ are hyperparameters.

**Training details.** We use Diffusion Policy's convolutional U-Net [52] with DDIM inference [53]. The model is conditioned on the fused visuo-tactile embedding and two consecutive proprioceptive observations. All encoder components—including the CLIP backbone, tactile CNN, and cross-attention layers—are finetuned during this stage using a learning rate of $3 \times 10^{-5}$.

## 5 Experiments

In this section, we address key questions about the role of touch in fine-grained manipulation and the influence of different pretraining strategies on downstream performance. Specifically, we ask: (1) How does touch enable fine-grained manipulation? (2) How does visuo-tactile pretraining improve policy robustness? (3) How do pretraining variations affect task performance?

### 5.1 Large-Scale Visuo-Tactile Data and Pretraining

To enable effective visuo-tactile pretraining, we curated a diverse dataset of over 2.6 million visuo-tactile pairs collected from 12 indoor and outdoor environments. This dataset comprises more than 2,700 demonstrations and spans 43 manipulation tasks. We categorize the data into three groups: (1) the four core tasks presented in this paper, (2) other indoor tasks that enhance distributional diversity, and (3) over 30 in-the-wild tasks designed to capture complex, real-world scenarios. The distribution of demonstrations across these three categories is shown in Fig. 4.

We evaluate the learned visuo-tactile representations through two complementary qualitative analyses. First, we test cross-modal reconstruction by providing partially masked tactile and RGB inputs and measuring the recovery of missing tactile signals, assessing whether the model captures meaningful visuo-tactile associations in both in- and out-of-distribution settings. Second, we visualize self-attention maps from the final ViT layer to examine whether the model consistently attends to contact-relevant regions in RGB images across diverse scenarios. Fig. 5 shows qualitative results from four representative tasks, including two in-distribution and two out-of-distribution examples.

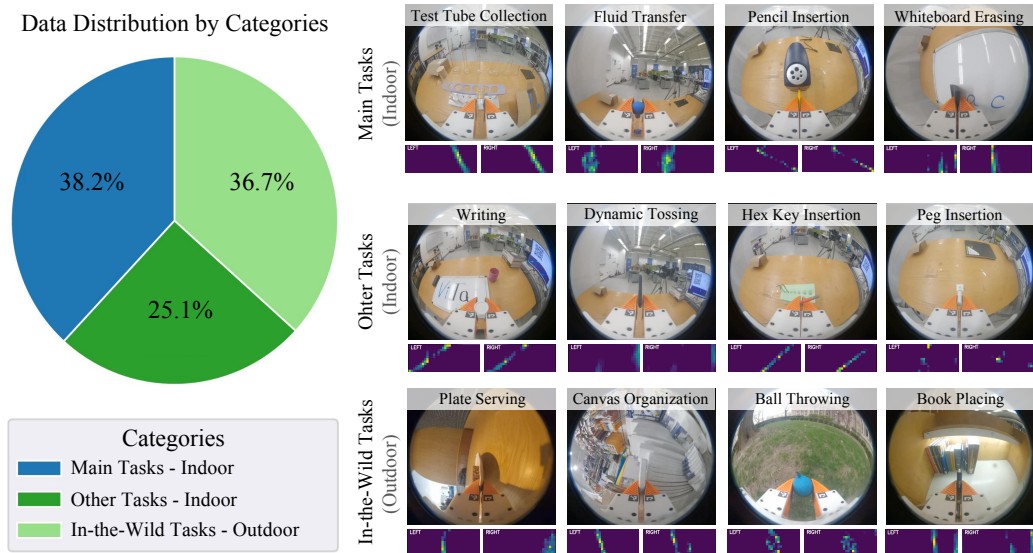

Figure 4: **Pretraining Data Distribution.** Our dataset comprises over 2,700 demonstrations, split across three categories: (1) the four core tasks introduced in this paper, (2) other indoor tasks to broaden the data distribution, and (3) in-the-wild tasks collected in diverse outdoor environments. We include representative examples from each category to highlight the variety in both task complexity and environmental context.

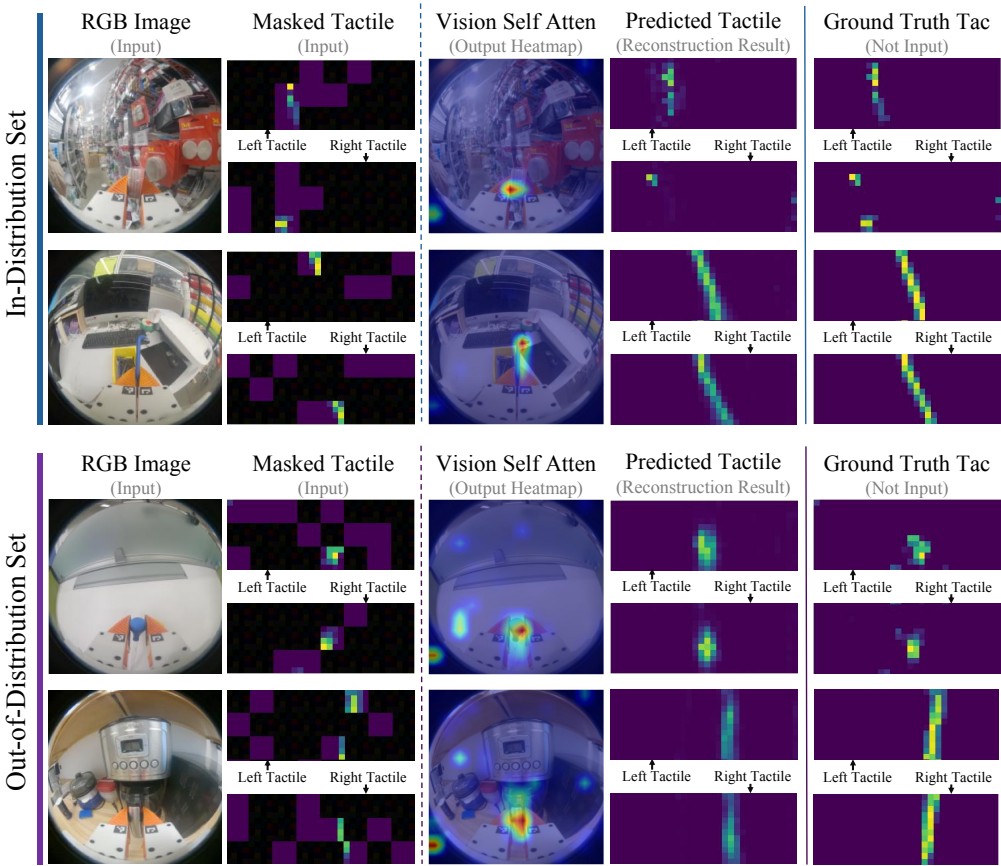

Figure 5: **Qualitative Results of Pretraining.** We show four examples illustrating our pretrained encoder's tactile reconstruction performance and ViT self-attention heatmaps. The encoder accurately reconstructs tactile images for both in- and out-of-distribution inputs, while the vision module consistently attends to the gripper–contact region, independent of background or object familiarity.

## 5.2 Experimental Setup

We evaluate our multi-modal sensing and learning system on four challenging real-world robotic tasks. Representative snapshots of each task are shown in Fig. 6. Below are the basic descriptions and evaluation metrics for all tasks:

*(1) Task Requiring In-Hand State Information*

**Test Tube Collection.** The robot must pick up a test tube from a box, reorient it in-hand using the test tube rack, and precisely insert it into the test tube rack. *Evaluation Metric:* The task is considered successful if the test tube is fully inserted into the test tube rack without being dropped or broken.

**Pencil Insertion.** The robot needs to insert a pencil into a sharpener. Since the pencil is initially grasped upright, the robot must first reorient it before performing a precise insertion. *Evaluation Metric:* The task is considered successful if the pencil is accurately inserted into the sharpener.

*(2) Task Requiring Fine-Grained Force Information*

**Fluid Transfer.** The robot uses a pipette to transfer fluid between containers, requiring precise pressure to extract liquid without dropping it. It then moves above the target container and gently squeezes to release the fluid. This task demands fine-grained force control. *Evaluation Metric:* The task is considered successful if the fluid is transferred to the second container without spilling.

**Whiteboard Erasing.** The robot uses a soft eraser to remove two strokes of text from the whiteboard. It must apply the right amount of pressure to erase the marker ink without exceeding force limits that could damage the system. The task requires consistent and controlled force throughout. *Evaluation Metric:* The task is considered successful if all visible marker ink is removed from the whiteboard.

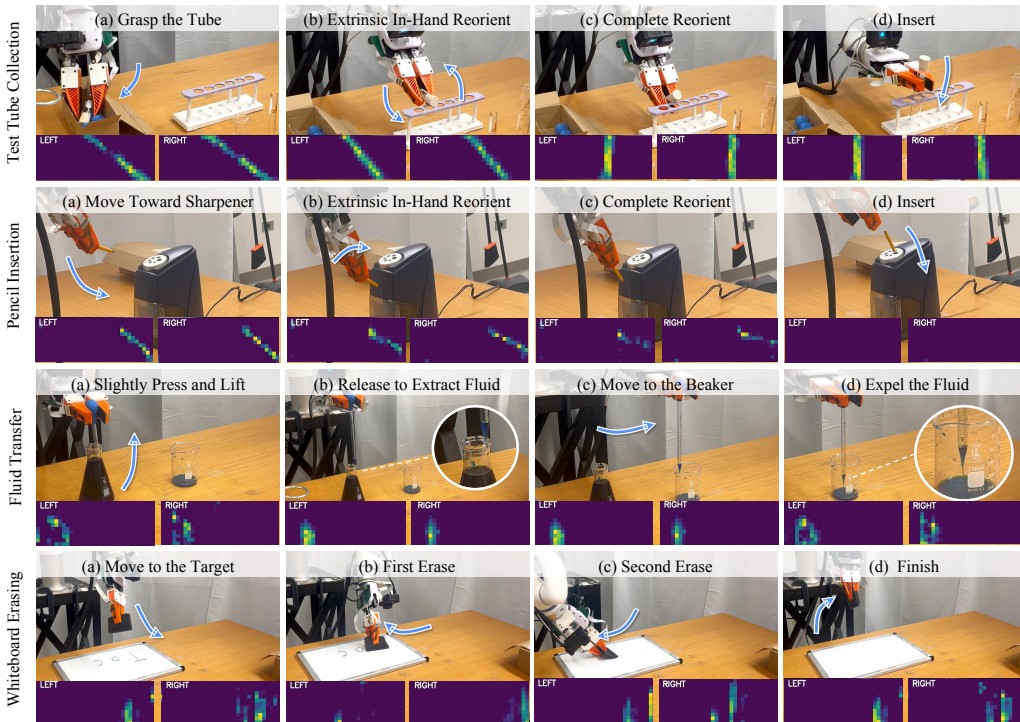

Figure 6: **Quantitative Results.** We evaluate our visuo-tactile policy across four fine-grained manipulation tasks. Descriptions and metrics of the tasks can be found in Sec. 5.

In the experiments, we compare our methods with the following baselines.

*(1) Vision-Only.* This method feeds one RGB image as input to CLIP, and extracts a 768-dimensional CLIP embedding. This embedding, along with the robot's proprioceptive information, is then fed into the image-based diffusion policy. We follow the same implementation as outlined in Chi et al. [24].

*(2) Ours w/o Cross-Attention.* This method does not employ any cross-attention between vision and touch. Instead, it processes two tactile images (from the left and right fingers) through a 3-layer CNN to produce a 512-dimensional feature vector, which is then simply concatenated with the visual embedding and proprioceptive states as input to the diffusion policy.

*(3) Ours w/o Pretraining.* This method uses the visuo-tactile encoder proposed in the paper, with the vision backbone initialized from CLIP and the other parts of the joint encoder initialized from scratch (no visuo-tactile pretraining). The resulting embedding is concatenated with proprioceptive inputs to condition the diffusion policy.

*(4) Ours w/ Pretraining.* This method uses the visuo-tactile encoder proposed in the paper, with weights obtained from visuo-tactile pretraining. Similarly, the output embedding from the visuo-tactile encoder is concatenated with robot proprioception as conditioning for the diffusion policy.

For each of the four tasks, we conduct 20 trials with moderate randomization of the initial robot position and environmental conditions. All policies are trained for 60 epochs, at which point they have converged. The results are presented in Table 1.

| Tasks Requiring In-Hand State Information | | | | | | | |
|---|---|---|---|---|---|---|---|
| **Modalities** | **Test Tube Collection (197 demos)** | | | | **Pencil Insertion (170 demos)** | | |
| | **Grasp** | **Reorient** | **Insert** | **Whole Task** | **Reorient** | **Insert** | **Whole Task** |
| Vision-Only | **1.00** | 0.25 | 0.25 | 0.25 | 0.50 | 0.65 | 0.45 |
| Ours w/o Cross-Attention | **1.00** | **1.00** | 0.50 | 0.50 | 0.80 | 0.75 | 0.70 |
| Ours w/o Pretraining | **1.00** | **1.00** | 0.70 | 0.70 | 0.80 | 0.80 | 0.75 |
| **Ours w/ Pretraining** | **1.00** | **1.00** | **0.85** | **0.85** | **0.95** | **0.90** | **0.85** |

| Tasks Requiring Fine-Grained Force Information | | | | | | | |
|---|---|---|---|---|---|---|---|
| **Modalities** | **Fluid Transfer (160 demos)** | | | | **Whiteboard Erasing (105 demos)** | | |
| | **Acquire** | **Transfer** | **Expel** | **Whole Task** | **First Erase** | **Second Erase** | **Whole Task** |
| Vision-Only | 0.95 | 0.85 | 0.55 | 0.55 | 0.65 | 0.65 | 0.55 |
| Ours w/o Cross-Attention | 0.90 | 0.75 | 0.75 | 0.70 | **0.70** | 0.50 | 0.45 |
| Ours w/o Pretraining | **1.00** | 0.90 | 0.80 | 0.80 | 0.60 | **0.75** | 0.60 |
| **Ours w/ Pretraining** | **1.00** | **1.00** | **0.90** | **0.90** | **0.70** | **0.75** | **0.70** |

Table 1: **Comparison with Baselines.** We evaluate our policy over 20 episodes and the best performance for each task is bolded. The numbers in parentheses indicate the number of training demonstrations.

## 5.3 Analysis and Discussion

Our system enhances a handheld gripper by integrating tactile sensing and training a large-scale visuo-tactile encoder to further improve manipulation policies. We observe three key benefits from incorporating touch and leveraging pretrained representations.

(1) *Tactile feedback provides explicit in-hand state information.* In a single-camera setup, visual inputs often suffer from severe occlusions. For example, in the Test Tube Collection task, the vision-only policy relied heavily on the color of the wooden cork to infer orientation. A minor change (such as switching to a lighter cork with less distinct features) confused the vision policy and degraded performance in reorientation. The tactile policy, however, remained unaffected by such variations.

(2) *Tactile feedback improves detection of critical state transitions.* In fine-grained, force-controlled tasks like Fluid Transfer, recognizing transitions between action phases is crucial. The vision-only policy struggles because visual cues, such as gripper width, appear similar before and after contact, making it hard to tell whether squeezing has ended. Consequently, it often skips the "expel water" phase prematurely. In contrast, the tactile policy uses pressure feedback to sense subtle force changes, accurately detecting when the water is fully released. This tactile awareness enables smoother phase transitions and improves both accuracy and robustness.

(3) *Joint visuo-tactile encoders enable more coordinated use of vision and touch.* Naive fusion methods that simply concatenate visual and tactile features, such as the policy without cross-attention, often fail to integrate the two modalities effectively, leading the policy to over-rely on one and neglect the other. This imbalance was evident in the Whiteboard Erasing task, where the policy without cross-attention applied excessive force to amplify tactile signals, triggering safety stops. In contrast,

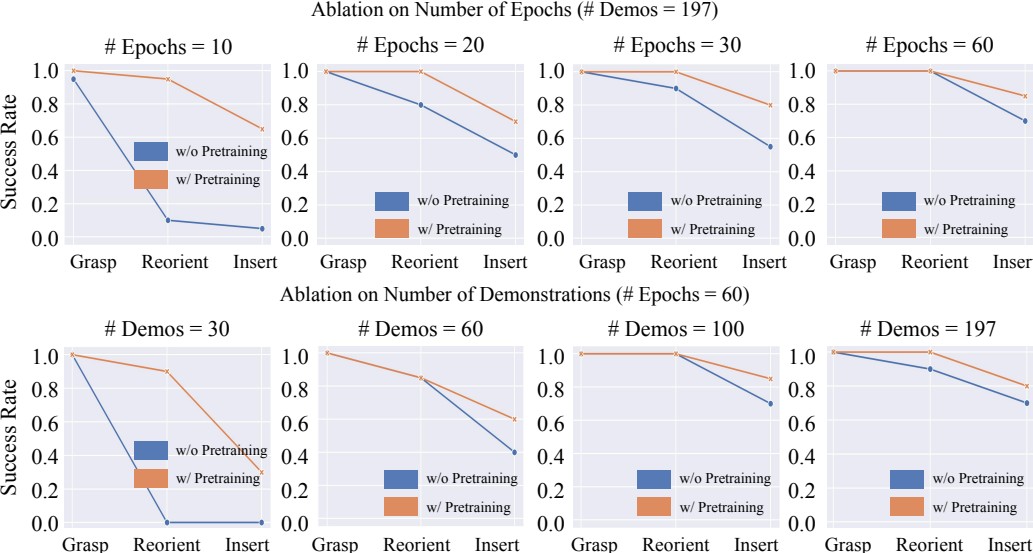

Figure 7: **Ablation Results for Varying Numbers of Training Demonstrations and Epochs for Test Tube Collection Task.** Our findings show that the policy with visuo-tactile pretraining consistently outperforms the policy without pretraining, both in low-epoch and low-demonstration regimes.

our jointly trained visuo-tactile encoder coordinates both modalities, enabling force modulation based on visual context and tactile feedback, and reducing failures caused by misuse of either modality.

**Pretraining Ablations: Varying Number of Training Demonstrations and Epochs.**

To assess pretraining effectiveness, we evaluate Test Tube Collection performance across varying numbers of demonstrations and training epochs, as shown in Fig. 7. We found that pretraining provides significant benefits, particularly in low-data and low-epoch training settings.

*(1) Low-Data Regime (Fewer Than 60 Demonstrations).* When training with only 30 or 60 demonstrations, the policy without pretraining often becomes stuck after grasping, uncertain how to proceed. In contrast, the policy with pretraining—even with just 30 demonstrations—follows much smoother trajectories and usually only fails during the final insertion step. We believe that pretraining helps the encoder to learn visuo-tactile correlations early on, which enables the downstream policy to focus on learning effective action trajectories.

*(2) Low-Epoch Regime (Fewer Than 60 Epochs).* In the low-epoch regime, we observed that the policy without pretraining was more sensitive to initial environmental configurations. For instance, when the test tube was placed at a steep incline and was difficult to grasp, an imperfect grasp position had a considerable impact on the execution of the reorientation task. The policy without pretraining sometimes over- or under-reoriented, resulting in failure. We believe that in the low-epoch regime, the policy with pretraining benefits from prior knowledge that emphasizes tactile cues, which makes it more robust to environmental disturbances.

# 6 Conclusion and Limitations

In this work, we present a handheld gripper enhanced with tactile sensing and introduce a large-scale visuo-tactile dataset. We demonstrate its utility by pretraining a visuo-tactile joint encoder and evaluating it on several fine-grained manipulation tasks using a single-arm robot equipped with a parallel gripper, mirroring the handheld setup. Looking ahead, we aim to extend this approach to multi-finger dexterous hands, where tactile feedback can enable even richer and more dexterous manipulation skills.

## Acknowledgement

This work is partially supported by the DARPA TIAMAT program (HR0011-24-9-0430), NSF Award #2409661, Toyota Research Institute (TRI), Sony Group Corporation, Samsung Research America (SRA), Google, Dalus AI, Pickle Robot, and an Amazon Research Award (Fall 2024). This article solely reflects the opinions and conclusions of its authors and should not be interpreted as necessarily representing the official policies, either expressed or implied, of the sponsors.

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

# Supplementary Materials

# Contents

## A    Cross-Sensor Generalization and Consistency

Throughout the project, the visuo-tactile gripper and robot grippers used different pairs of tactile sensors, providing an inherent test of cross-sensor generalization. To further evaluate sensor consistency, we replaced the robot's tactile sensors with newly fabricated ones that had not been used previously in this project and re-evaluated the Test Tube Collection task with the new sensors. As shown in Table 2, the success rates across all subtasks remained nearly identical, showing strong cross-sensor consistency and confirming that the sensors are reliable and interchangeable for downstream deployment.

| Test Tube Collection Task | Grasp | Reorient | Insert | Whole Task |
|---|---|---|---|---|
| Policy w/ Pretraining (Original Sensors) | 1.00 | 1.00 | 0.85 | 0.85 |
| Policy w/ Pretraining (New Sensors) | 1.00 | 1.00 | 0.85 | 0.85 |

Table 2: Policy performance with new tactile sensors. Consistent downstream performance confirms cross-sensor reliability.

# B  Pretraining Ablations

## B.1  Effect of Demonstration Quantity on Pretraining

To assess the impact of pretraining data, we analyze how the size of the pretraining dataset influences reconstruction and representation quality in our visuo-tactile encoder. As shown in Fig. 8, larger datasets yield lower reconstruction losses and more stable training. Corresponding tactile reconstructions and ViT self-attention maps show that with more data, attention concentrates more precisely on gripper contact regions, and reconstructions become cleaner and more structured, which signals stronger cross-modal representations.

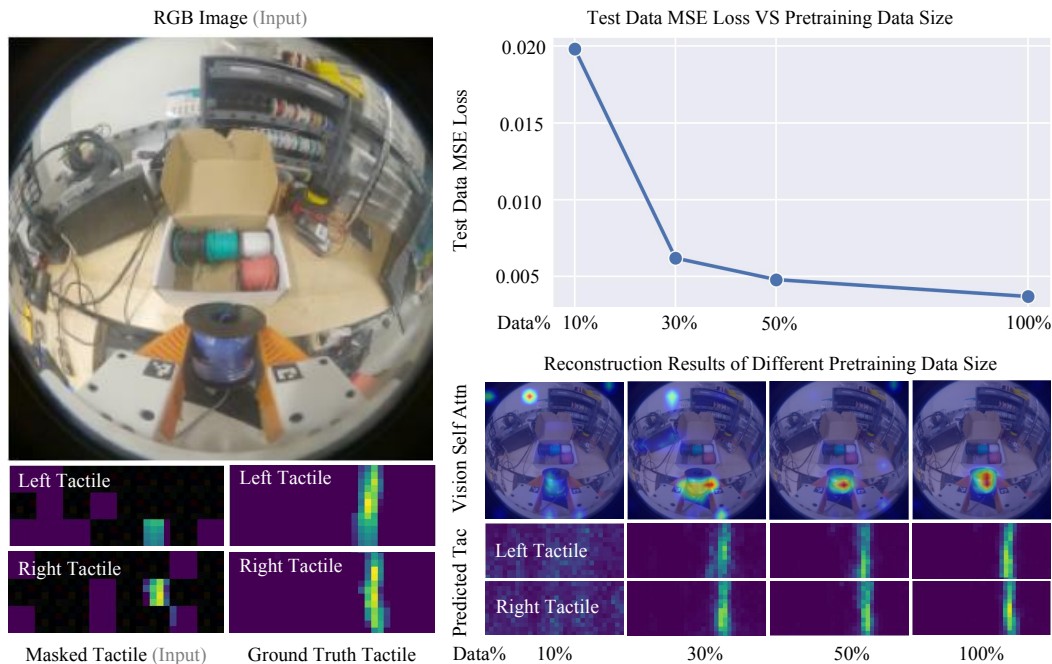

Figure 8: **Impact of Pretraining Dataset Size on Cross-Modal Reconstruction.** On the left, we show a representative visuo-tactile example with RGB image, the masked tactile input, and the ground-truth tactile image. On the right, the top panel plots test data MSE loss against the amount of pretraining demonstrations, and the bottom panel displays reconstructed tactile images alongside their visual self-attention heatmaps for four different dataset amounts. As the amount of pretraining data grows, the test MSE steadily decreases and the reconstructions become notably cleaner and more accurate.

## B.2  Impact of Removing Task-Specific Demonstrations on Pretraining

To evaluate the robustness and generalization of our visuo-tactile pretraining, we conducted two experiments where we removed demonstrations for the specific downstream fine-tuning tasks from the pretraining dataset. In the first experiment, all demonstrations for the Test Tube Collection task were excluded, and in the second, all demonstrations for the Whiteboard Erasing task were removed. This setup allowed us to assess how effectively the pretrained visuo-tactile encoder generalizes when trained solely on in-the-wild data, without any task-specific examples included during pretraining.

After each removal, we retrained the encoder on the reduced dataset, fine-tuned a downstream policy for the corresponding task, and then evaluated its real-world performance. The resulting pretraining dataset contained approximately 2,500 in-the-wild demonstration videos, compared to the original 2,700.

Qualitatively, across both held-out tasks, we observed that the ViT attention heatmaps, evaluated on both training and held-out tasks, remained focused on the gripper's contact regions, consistent with those obtained using the full pretraining dataset. This indicates that the encoder continued to

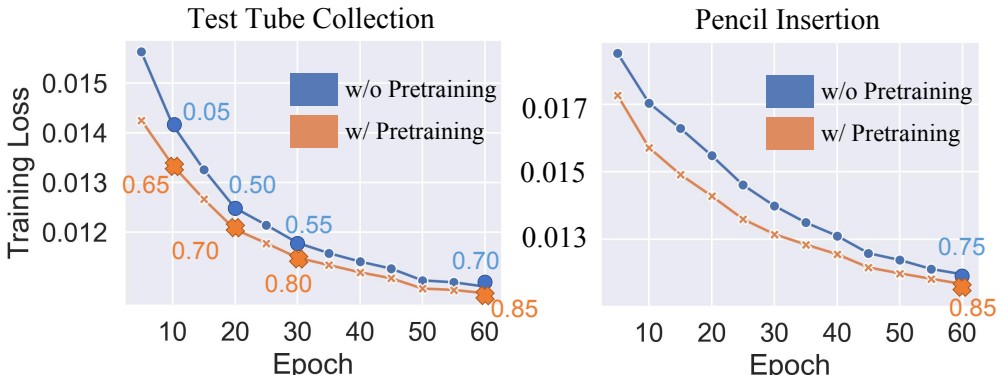

Figure 9: **Downstream Policy Training Loss for Two Tasks.** Training loss curves from epoch 5 to 60 for Test Tube Collection and Pencil Insertion tasks. Numbers along each curve indicate rollout success rates at corresponding epochs. Policies with pretraining show faster convergence and consistently higher success rates compared to policies without pretraining.

capture meaningful visuo-tactile correspondences even when the downstream tasks were excluded from pretraining.

Quantitatively, as shown in Table 3, the downstream policies trained with and without the corresponding task data in pretraining achieved nearly identical performance. These results demonstrate that the learned visuo-tactile representations generalize effectively to unseen manipulation tasks rather than overfitting to the specific demonstrations present in the pretraining data.

Effect of Removing Task-Specific Demonstrations From Pretraining

| Policy | Test Tube Collection | | | | Whiteboard Erasing | | |
|---|---|---|---|---|---|---|---|
| | **Grasp** | **Reorient** | **Insert** | **Whole Task** | **First Erase** | **Second Erase** | **Whole Task** |
| With Pretraining (Ours) | **1.00** | **1.00** | **0.85** | **0.85** | 0.70 | **0.75** | 0.70 |
| Without Task Data in Pretraining | **1.00** | **1.00** | 0.80 | 0.80 | **0.75** | **0.75** | **0.75** |

Table 3: Downstream policy performance when task-specific demonstrations were excluded from pretraining. We removed all demonstrations for (1) Test Tube Collection and (2) Whiteboard Erasing from the pretraining dataset. The resulting performance remains comparable to that achieved with the full dataset, indicating that the pretrained visuo-tactile encoder generalizes well even without task-specific data.

## C  Downstream Task Training Analysis

### C.1  Impact of Pretraining on Training Loss Convergence

To evaluate the impact of visuo-tactile pretraining on downstream task performance, we analyze both task success rates and training convergence behavior. In addition to final success rates for the Test Tube Collection task under low-demonstration and low-epoch regimes with and without pretraining (see section 5 in the main paper), we also track training loss curves for two representative tasks—Test Tube Collection and Pencil Insertion—with and without pretraining, as shown in Fig. 9.

We plot training loss from epoch 5 to epoch 60 for both settings. Across both tasks, we observe that policies with visuo-tactile pretraining consistently achieve lower loss values throughout training compared to those without pretraining. This indicates faster convergence and improved training stability.

Importantly, even when policies trained with and without pretraining converge to similar loss levels during training, we observe that policies with pretraining consistently achieve higher success rates in real-world rollouts. We attribute this discrepancy to the difference between open-loop training (measured by loss) and closed-loop execution (evaluated via physical rollouts). In real robot experiments, downstream tasks are subject to unpredictable variations in object positions, environmental dynamics, and contact conditions. Policies with visuo-tactile pretraining are more robust to these variations

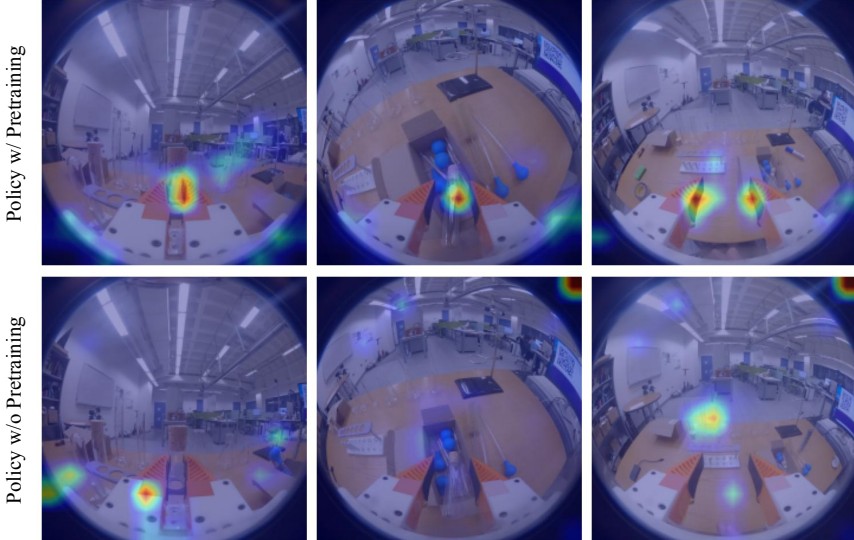

Figure 10: **ViT Self-Attention Maps after Downstream Fine-tuning.** The self-attention maps from the ViT module show that policies with pretraining tend to focus on task-relevant contact regions, such as the gripper and object. In contrast, policies without pretraining often attend to background features, like table edges or box boundaries, resulting in less stable downstream performance.

because they have learned to rely on tactile cues that generalize across scenarios. As a result, they exhibit more reliable and consistent behaviors, particularly in out-of-distribution conditions.

### C.2 Policy Self-Attention Maps Before vs. After Downstream Fine-Tuning

To further understand the performance gap between policies with and without pretraining observed during real-world evaluation, we analyzed the self-attention heatmaps from the ViT module in the Test Tube Collection task, as shown in Fig. 10. The policy with pretraining consistently focuses attention on the gripper-object contact region, suggesting that it retains spatial priors critical for manipulation-priors learned during large-scale visuo-tactile pretraining. In contrast, the policy without pretraining frequently directs attention to irrelevant background features, such as table edges or nearby objects, likely in an attempt to infer contact indirectly. This misplaced focus can degrade performance, particularly in cluttered or visually diverse environments. These attention patterns provide insight into why the policy with pretraining not only learns more efficiently but also generalizes better across task variations.

## D  Baseline Failure Analysis

To better understand the challenges involved in fine-grained real-world manipulation, we analyze common failure cases across baseline policies for the four core tasks. Fig. 11 presents representative examples of failures for both vision-only policies and policies without cross-attention. These cases highlight the limitations of relying solely on visual or static tactile input and emphasize the importance of integrated visuo-tactile feedback.

**(a) Test Tube Collection.** In the Test Tube Collection task, vision-only policies often struggle to differentiate between the reorientation and insertion phases. As shown in Fig. 11(a), a typical failure involves prolonged hesitation when deciding whether to continue reorienting or to proceed with insertion. Even after a successful reorientation, the policy may continue the motion unnecessarily, suggesting uncertainty about the current phase of the task.

Although policy without cross-attention transitions more decisively between phases, it can become overly dependent on localized tactile cues. This sometimes results in the robot getting stuck during

Typical Failure Cases of Baselines

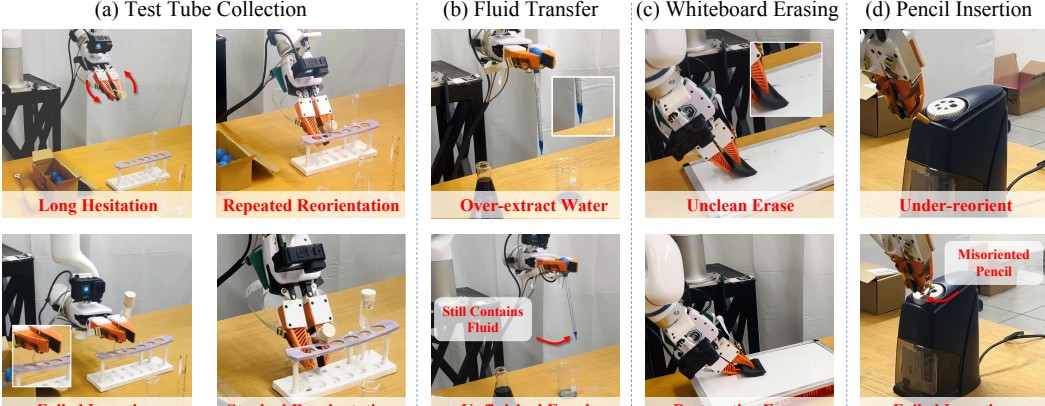

Figure 11: **Failure Cases.** We present typical failure cases of baseline methods for all four tasks and analyze the reasons for these failures to highlight the complexity of the tasks and the importance of tactile feedback and pretraining during these steps.

reorientation, where it maintains excessive contact without progressing. Additionally, the policy without cross-attention frequently fails during insertion, likely due to insufficient use of visual input for aligning the test tube with the narrow slots of the test tube rack.

**(b) Fluid Transfer.** Fluid Transfer requires precise, force-sensitive control—something vision-only policies typically struggle to provide. As illustrated in Fig. 11(b), a common failure during fluid extraction involves excessive squeezing, where the robot either applies too much pressure or continues to squeeze for too long. Without tactile feedback, the policy relies entirely on ambiguous visual cues, often resulting in drawing too much fluid. This overfilling can hinder subsequent steps, particularly during the expel fluid phase, where the robot may fail to fully release the fluid. The result is an imprecise release and overall task failure.

Failures are also frequent during the expel fluid phase. The policy may skip the expel fluid step entirely, particularly when the fluid is not clearly visible—either due to an insufficient amount drawn or when the second beaker contains nearly no water. These behaviors suggest that vision-only policy is overly sensitive to subtle environmental variations and fails to detect critical task-related cues.

**(c) Whiteboard Erasing.** Whiteboard Erasing requires consistent force against a flat surface. Vision-only policies often fail due to incomplete visual feedback regarding contact between the eraser and the board. As a result, insufficient pressure is applied, leaving visible marks and leading to task failure (Fig. 11(c)).

On the other hand, policies without cross-attention tend to overcompensate by pressing too hard in an attempt to maximize tactile feedback. This often causes the robot to apply excessive force, triggering safety stops and prematurely terminating the task.

**(d) Pencil Insertion.** Pencil Insertion requires precise in-hand reorientation of the pencil before the final insertion. Vision-only policies often struggle with this step, particularly when the pencil is held low in the gripper, partially occluded, or only slightly inclined (Fig. 11(d)). These conditions limit the visibility of the pencil and lead to inaccurate orientation estimates. As a result, the policy frequently misjudges the pencil's orientation, causing insertion failures.

# E  Training Hyperparameters for Pretraining

| Parameter | Value | Parameter | Value |
|---|---|---|---|
| Seed | 42 | Batch size | 128 |
| Tactile embed dimension | 768 | Epochs | 5 |
| Tactile patch size | 4 | CLIP learning rate | $3 \times 10^{-5}$ |
| Number of attention heads | 8 | Encoder learning rate | $1 \times 10^{-4}$ |
| Cross-attention dropout | 0.20 | Weight decay | $2 \times 10^{-3}$ |
| Decoder hidden dim | 768 | Warmup ratio | 0.10 |
| Scheduler | Linear warmup + Cosine annealing | Optimizer | AdamW |
| Validation ratio | 0.10 | Tactile masking ratio | 60%–80% |

Table 4: Pretraining hyperparameters and training configuration. Tactile masking ratio means random patch masking during pretraining.

# F  Detailed Hyperparameters for Downstream Task Training

| Observation Settings | | Optimization Parameters | | | |
|---|---|---|---|---|---|
| Image obs. horizon | 2 | Optimizer | AdamW | Momentum | $\beta_1 = 0.95, \beta_2 = 0.999$ |
| Proprio. obs. horizon | 2 | LR (action diff.) | $3 \times 10^{-4}$ | LR (pretrained enc.) | $3 \times 10^{-5}$ |
| Action horizon | 16 | LR schedule | Cosine decay | Batch size | 64 |
| Obs. resolution | $224 \times 224$ | Train diff. steps | 50 | Inference steps | 16 |

Table 5: Training hyperparameters for policy learning. Observation settings on the left; optimization parameters on the right.

# G  Extended Task Descriptions

In this section, we provide detailed information on the four tasks described in the main paper. We describe the motions and evaluation metrics for each step, highlighting how these steps demonstrate the capabilities of our tactile sensors.

## G.1  Details for the Test Tube Collection Task

*Initial Positions.* At the start of each episode, the robot is positioned near a cluttered box containing a test tube. The test tube is partially visible, with its close end typically protruding slightly outside the box, while the rest is embedded among other objects. Several aspects are randomized across episodes: the robot's initial pose, the position and orientation of the test tube, the clutter surrounding the test tube, and the color of the wooden cork. These variations introduce uncertainty in the grasping setup and require precise spatial reasoning and tactile feedback for reliable performance.

*Step 1: Grasp the Test Tube.* The robot reaches into the box to grasp the partially exposed test tube. Depending on its orientation and surrounding clutter, the robot may need to adapt to a wide range of grasping conditions. This step demonstrates the strength of our thin, flexible tactile sensor in enabling precise manipulation within narrow and constrained environments. *Evaluation Metrics:* The step is considered successful if the robot securely grasps the test tube and prepares to proceed to the next step.

*Step 2: Reorient the Test Tube.* After grasping, the robot moves to the test tube rack and uses its edge to rotate the test tube by approximately 70 degrees. This reorientation is particularly challenging due to the test tube's transparency, which can cause vision-only policies to misjudge in-hand orientation and either under- or over-rotate. In contrast, tactile feedback provides clear contact signals, enabling reliable reorientation. *Evaluation Metrics:* The step is successful if the robot completes a single,

correct reorientation. The task fails if multiple attempts are required or if the final orientation remains incorrect.

*Step 3: Insert the Test Tube.* Finally, the robot attempts to insert the reoriented test tube into a tight-fitting slot on the rack. Given the transparent material and narrow tolerance, accurate alignment is critical. Tactile sensing plays a key role in localizing the body of the test tube and guiding the insertion process. *Evaluation Metrics:* The step is considered successful if the test tube is inserted fully and securely into the rack.

## G.2 Details for the Fluid Transfer Task

*Initial Positions.* The robot begins with its parallel grippers positioned around the bulb of a pipette, roughly centered between its fingers. Across episodes, several aspects are randomized: the robot's initial pose, the orientation and position of the pipette within the first beaker, the location of both beakers, the amount of water in each beaker, and the water's color and opacity. These variations introduce uncertainty in grasping and perception, making tactile sensing essential for accurately locating the bulb and applying the correct amount of force.

*Step 1: Grasp Pipette and Acquire Fluid.* The robot first applies gentle pressure to the pipette's bulb to extract water from the first beaker. This step requires fine control of squeezing force—too little and not enough water is drawn, too much and the pipette may slip or spill. The robot must also detect when a sufficient amount of liquid has been collected, and then lift the pipette while maintaining a stable grasp, preparing for the transfer. Visual feedback is unreliable here due to partial occlusion of the bulb and the small volume of liquid involved. Our tactile sensors provide accurate contact localization and force feedback, enabling the robot to perform this motion precisely and without spillage. *Evaluation Metrics:* The step is considered successful if the robot extracts an appropriate amount of liquid and lifts the pipette without spilling.

*Step 2: Transfer to the Second Beaker.* With the pipette securely grasped and filled, the robot moves toward the second beaker. It must maintain consistent pressure on the bulb throughout the motion to avoid accidental release or dropping the pipette. This step requires precise motion planning and continuous force regulation, as even small deviations in grip force can cause the fluid to spill. Tactile sensing ensures grip stability, while vision assists in coarse positioning. *Evaluation Metrics:* The step is successful if the robot moves the pipette directly above the second beaker without spilling any water during the transfer process.

*Step 3: Expel Fluid.* Once positioned above the second beaker, the robot lowers the pipette and squeezes the bulb to release the water. This step requires accurately detecting contact and applying the right amount of force to fully expel the liquid. Vision-only policies often struggle in this scenario due to the transparency of both the pipette and the fluid, making it difficult to verify whether the liquid has been released. In contrast, our visuo-tactile policy leverages force signals to monitor the release process and prevent premature lifting. *Evaluation Metrics:* The step is considered successful if all the liquid is transferred to the second beaker and the pipette is lifted cleanly, with no remaining water or unintended spills.

## G.3 Details for the Whiteboard Erasing Task

*Initial Positions.* The robot starts with a soft foam brush held vertically in its gripper. Across episodes, several aspects of the setup are randomized: the robot's initial pose, the height and position of the whiteboard, and the exact location of the brush within the gripper. These variations introduce uncertainty in alignment and contact, making tactile sensing critical for successful execution.

*Step 1: Erase First Stroke.* The robot begins with the foam brush held vertically in its gripper. To prepare for erasing, it first rotates its gripper by approximately 90 degrees, reorienting the brush into a horizontal position. It then moves toward the whiteboard and attempts to erase the letter "T" using a vertical stroke. The robot must apply a slight downward force and rely on tactile feedback to confirm contact with the whiteboard surface. This step is challenging because visual input alone

may not reliably indicate whether the brush has made effective contact, especially in the presence of occlusions or slight misalignments. *Evaluation Metrics:* The step is considered successful if the robot fully erases the letter "T" with proper contact and without slipping or losing control of the brush.

*Step 2: Erase Second Stroke.* After completing the vertical stroke, the robot reorients the brush to a vertical position and then performs a horizontal stroke to erase the letters "ac." Tactile sensing is again crucial—not only for confirming contact with the surface, but also for detecting when sufficient pressure is applied to begin effective erasure. *Evaluation Metrics:* The step is considered successful if the robot cleanly erases the letters "ac" while maintaining stable and continuous control of the brush.

### G.4 Details for the Pencil Insertion Task

*Initial Positions.* The robot begins with each episode holding a slightly inclined pencil in its gripper. Several factors are randomized across episodes: the pencil's orientation and position within the gripper, the placement and height of the pencil sharpener, the size and alignment of the insertion hole, and the robot's initial pose. These variations introduce uncertainty that requires fine-grained control and perception.

*Step 1: Reorient the Pencil.* The robot first moves toward the vertical surface of the pencil sharpener and uses it as a reference to reorient the pencil. The goal is to align the pencil so that it becomes parallel to the gripper and ready for insertion. This step is challenging due to the pencil's small size and partial occlusion within the gripper, which makes it difficult to estimate its pose using vision alone. Tactile feedback provides critical information for detecting contact and adjusting alignment. *Evaluation Metrics:* This step is considered successful if the pencil is reoriented to be parallel to the robot's gripper, ensuring it is properly aligned for insertion.

*Step 2: Insert the Pencil.* With the pencil correctly oriented, the robot must insert it into the top hole of the pencil sharpener. This step demands precise positioning, fine alignment, and controlled force to avoid missing the hole or applying excessive pressure, which could cause the pencil to slip or become jammed. Tactile sensing enables the robot to detect contact with the hole and adjust the insertion motion accordingly. *Evaluation Metrics:* The step is considered successful if the pencil is inserted smoothly into the sharpener without slipping, jamming, or needing realignment.

