# OpenReview forum: "Touch in the Wild: Learning Fine-Grained Manipulation with a Portable Visuo-Tactile Gripper"
_NeurIPS.cc/2025/Conference — NeurIPS 2025 poster_

### Official Review · Reviewer_meGV · 2025-07-02

**Clarity:** 3
**Significance:** 1
**Originality:** 1
**Rating:** 4
**Confidence:** 5

**Summary:**

In this paper, the authors first present a data collection hardware for multi-modal data collection, which integrates RGB camera and tactile sensors. Then  they designs a cross-modal representation learning framework to intergrate these modalties through pretraining. The authors evaluate their hardware and method on 4 real-world manipulation tasks, where it outperforms the vision-only baseline. Ablation studies demonstrate the effectiveness of multi-modal pretraining.

**Questions:**

1. Initial state distribution: The consistency of the initial states of objects across the comparisons is unclear. The authors should explicitly state whether the initial conditions are standardized to ensure fair comparision.

2. Comparision with other multi-modal pretraining methods should be added.

3. The authors use a parallel gripper from UMI as the data collection system. However, it is deployed on XARM gripper, whose design is different. The paper should explain why this discrepancy does not affect performance.

4. The inclusion of an ablation study analyzing the impact of in-the-wild data would strengthen the paper.

5. Tactile Sensor Consistency: The consistency of different tactile sensor instances is not addressed. The authors should clarify whether the same sensor instances are used during data collection and policy deployment.

**Ethical Concerns:**

["NO or VERY MINOR ethics concerns only"]

**Final Justification:**

Some of my concerns are addressed. But  the technical novelty does not meet the standard of NeurIPS. And the comparison with the work of Vitamin should be strengthened.

**Limitations:**

yes

**Quality:**

2

**Strengths And Weaknesses:**

Strengths:
1. A robot-free data collection system that can collect visual and tactile data simultaneously.
2. A multi-modality pretraining framework that improves training and sample efficiency.
3. The paper is well written,and the quality of visualizations is good.

Weakness:
1. Similarity: The paper structure, including the multi-modal pretraining, task design (tube reoriation, insertion) and ablation studies (w/o pretraining) , are very similar to https://arxiv.org/abs/2504.06156. While it is cited, it is not well described and discussed in the related work.

2. Technical novelty: The novelty of the hardware is very limited. It just intergrates the tactile sensor into UMI. The pretraining of multimodal representation is common and standard design in the community. No new architectures are introduced.

Dave V, Lygerakis F, Rueckert E. Multimodal visual-tactile representation learning through self-supervised contrastive pre-training[C]//2024 IEEE International Conference on Robotics and Automation (ICRA). IEEE, 2024: 8013-8020.

Kerr J, Huang H, Wilcox A, et al. Self-supervised visuo-tactile pretraining to locate and follow garment features[J]. arXiv preprint arXiv:2209.13042, 2022.

3. Title: It is named as "in the wild". but it still needs to a laptop to collect the tactile signals. So it is not a stand-alone system.

4. Experiments: While the authors collect data in the environments that are different from the tasks, its usefulness isnot validated in the experiment.

---

> ### Author Rebuttal · Authors · 2025-07-31
>
> >**Q1**: Similarity: The paper structure, including the multi-modal pretraining, task design (tube reoriation, insertion) and ablation studies (w/o pretraining) , are very similar to https://arxiv.org/abs/2504.06156. While it is cited, it is not well described and discussed in the related work.
>
> **A1**: While both projects share the high-level goal of enabling visuo-tactile learning for contact-rich manipulation using portable systems, there are several key differences that distinguish our work:
>
> a) Hardware and System Design:
>
> Our work goes beyond integrating tactile sensors into handheld devices. We deliberately chose a piezoresistive tactile sensor for large-scale in-the-wild data collection due to its unique advantages: (i) Lightweight. (ii) Easy integration with soft grippers. (iii) We also hope to highlight another key advantage that may not be obvious at first glance on the **sensor consistency**. We designed and fabricated custom flexible PCBs to ensure sensor consistency, enabling cross-sensor transfer. This means we can collect data with one pair of sensors and successfully rollout policies with another sensor pair. In contrast, the ViTaMIn, they use the same sensor pair, likewise many other optical-based tactile sensors, maintaining the consistency among different pairs of sensors is noticeably harder, limiting their scalability. Our design supports scaling to multiple data collection devices with different sensor pairs.
>
> b) Dataset:
>
> A core contribution of our work is large-scale visuo-tactile data collection in the wild, covering 43 diverse manipulation tasks. The dataset contributes unique value to the community in considering both vision and touch, and also physical interactions in the wild (not just touch as in paper [1]). We believe this scale and diversity largely advance the study of visuo-tactile correlation.
>
> [1] Tactile-Augmented Radiance Fields
>
> c) Downstream Task Design
>
> Regarding downstream tasks, although both our work and ViTaMIn include test tube reorientation, the task complexity differs substantially. In ViTaMIn, the test tube is grasped, reoriented, and the task concludes. In contrast, our setup involves grasping the test tube from within a cluttered container, performing reorientation, and then executing a precise insertion to verify the accuracy of reorientation. This added step demands a higher robustness of policy, making our task setting significantly more challenging.
>
> >**Q2**: Technical novelty: The novelty of the hardware is very limited. It just intergrates the tactile sensor into UMI. The pretraining of multimodal representation is common and standard design in the community. No new architectures are introduced.
>
> **A2**: a) Our work goes beyond integrating tactile sensors into handheld devices. See answer of **Q1**.
>
> b)  While our masked autoencoding approach builds on a standard framework, our architecture is carefully designed to meet the unique challenges of a single-camera, handheld gripper setup. This targeted design enables it to outperform baselines like VITaL and MVT (see **Q6**). We believe our approach offers meaningful insights for advancing visuo-tactile representation learning, particularly in the context of large-scale, in-the-wild datasets.
>
> >**Q3**: Title: It is named as "in the wild". but it still needs to a laptop to collect the tactile signals. So it is not a stand-alone system.
>
> **A3**: While our current setup uses a laptop for data logging, this component is not fundamental to the system design. It can be easily replaced with any lightweight, Ubuntu-based computing unit—a common approach in mobile robotic systems operating in the wild. Our focus is on the portable sensing and data collection hardware, which is fully compatible with embedded systems for stand-alone deployment.
>
> >**Q4**: Experiments: While the authors collect data in the environments that are different from the tasks, its usefulness isnot validated in the experiment.
>
> **A4**: To verify the usefulness of in-the-wild data, we removed all test tube collection tasks from the pretraining dataset and did additional experiments. This results in a total of ~2500 in-the-wild videos for pretraining (previously ~2700). We pretrained the encoder on this reduced dataset, then finetuned it on the test tube collection task. Finally, we evaluated policy performance.
>
> i) Qualitative Observations: The attention heatmap on both the training and validation set images remains primarily focused on the gripper region.
>
> ii) Quantitative Result: We conducted policy rollouts for the test tube collection task for both policies. We observed no behavioral inconsistencies between the two policies:
>
> | Task                                                       | Grasp | Reorient | Insert | Whole Task |
> |------------------------------------------------------------|-------|----------|--------|-------------|
> | Policy w/ pretraining (ours)                               | 1.0   | 1.0      | 0.85   | 0.85        |
> | Policy w/ pretraining (without tube in pretrain data)      | 1.0   | 1.0      | 0.80   | 0.80        |
>
> >**Q5**: Initial state distribution: The consistency of the initial states of objects across the comparisons is unclear. The authors should explicitly state whether the initial conditions are standardized to ensure fair comparision.
>
> **A5**: We ensured fair comparisons by standardizing initial conditions across all policy evaluations. All four policies were rolled out in the same session using an identical workspace setup. We also recorded and verified overlapping initial RGB frames to confirm state consistency.
>
> >**Q6**: Comparision with other multi-modal pretraining methods should be added.
>
> **A6**: As suggested by the reviewer 5R2Q,  we contact baseline multi-modal pretraining methods M2VTP and VITaL to our pipeline for a fair evaluation. Our method achieves higher success rate and leads to faster convergence and consistently reaches a lower final loss compared to baselines. The following tables show the downstream tasks success rate for ours and two baselines for test tube collection task:
>
> **Downstream Tasks Success Rate**
>
> | Task                                     | Grasp | Reorient | Insert | Whole Task |
> |------------------------------------------|-------|----------|--------|-------------|
> | MAE (ours)                                | 1.0   | 1.0      | 0.85   | 0.85        |
> | M2VTP                                      | 1.0   | 0.9      |  0.1   |   0.1      |
> | VITaL                                         | 0.1  | 0.0      | 0.0   | 0.0        |
>
> **Analysis**: Our analysis shows that these methods are tailored to fundamentally different use cases and are not well-suited to the demands of single-camera, in-the-wild manipulation, which is the focus of our work. Due to length limit, please refer to reviewer 5R2Q **Q2** for the detailed implementation and analysis.
>
> >**Q7**: The authors use a parallel gripper from UMI as the data collection system. However, it is deployed on XARM gripper, whose design is different. The paper should explain why this discrepancy does not affect performance.
>
>
> **A7**: We calibrate the gripper width by aligning the minimum and maximum openings of both grippers and applying linear interpolation. This mapping proved effective in practice. A similar approach was used in the original UMI paper (Chi et al.), where data collected with the UMI gripper was successfully deployed on the WSG gripper. This cross-gripper transfer is a standard and validated solution.
>
> >**Q8**: The inclusion of an ablation study analyzing the impact of in-the-wild data would strengthen the paper.
>
> **A8**: Please see our response to **Q4**
>
> >**Q9**: Tactile Sensor Consistency: The consistency of different tactile sensor instances is not addressed. The authors should clarify whether the same sensor instances are used during data collection and policy deployment.
>
> **A9**: All sensors were fabricated using the same manufacturing process based on flexible PCBs from factory, ensuring high consistency across instances. Thanks to this consistency, policies trained using data collected from one sensor pair can be reliably executed on a different sensor pair during deployment, without additional replacement. This cross-sensor feature is one of the key advantages of our system design for easy transfer cross embodiment and large scale data collection in the wild.
>
> Quantitative Results:
>
> We processed 60 Hz tactile data from training and recorded 10 Hz tactile signals across 20 successful rollouts. We then compared the statistical distributions of training and rollout signals, confirming their alignment.
>
> ### Per-pixel Distribution Statistics
>
> | Split | n       	| mean   | std	| min	| max	| skew   | kurt   |
> |-------|-------------|--------|--------|--------|--------|--------|--------|
> | Train (Human Demonstrations) | 286,114,560 | 0.0461 | 0.1699 | 0.0000 | 1.0000 | 3.9312 | 14.8065 |
> | Eval (Robot Rollouts) | 1,324,800   | 0.0414 | 0.1523 | 0.0000 | 1.0000 | 4.0988 | 16.8261 |
> —
>
> ### Per-episode Distribution Statistics
>
> | Split          	| mean   | std of means | median | 95%ile  |
> |--------------------|--------|---------------|--------|---------|
> | Train episode-means (Human Demonstrations) | 0.0461 | 0.0035    	| 0.0465 | 0.0512  |
> | Eval episode-means (Robot Rollouts) | 0.0414 | 0.0036    	| 0.0409 | 0.0477  |
>
> Results show similar tactile signal levels between train (mean = 0.046, std = 0.170) and eval (mean = 0.041, std = 0.152), with tight episode-level variability (train std = 0.0035; eval = 0.0036). Both distributions exhibit strong right skew and high kurtosis, indicating sparse background with high-intensity contacts—consistent with touch-driven behavior.

---

> > ### Comment · Reviewer_meGV · 2025-08-05
> >
> > Thank the authors for the detailed response. However, the technical novelty of the multi-modal representation learning (which is asked by other reviewers) is still not well addressed. The pre-training pipeline is same as existing works.
> >
> > While the difference between this work and Vitamin is explained, the main difference lies in the sensor, which is not original and was proposed in previous works.
> >
> > The effectiveness of the in-the-wild data is not significant according to the experiment results.
> >
> > Overall, i'll raise my score to `reject'

---

> > > ### Author Response · Authors · 2025-08-07
> > > **Official Comment by Authors**
> > >
> > > >**Q1**: However, the technical novelty of the multi-modal representation learning (which is asked by other reviewers) is still not well addressed. The pre-training pipeline is same as existing works.
> > >
> > >
> > >
> > > Thank you for the continued feedback. Regarding the contribution of our multi-modal representation learning method, we would appreciate it if you could point us to specific prior work you believe is important for comparison.
> > >
> > > In response to Reviewer 5R2Q’s suggestion, we have included comparisons with ViTAL and M2VTP. Our method significantly outperforms both, as shown in **Q6** of our earlier response. The improved performance stems from our pre-training method, which is specifically tailored to a single-camera, handheld gripper setup operating in the wild. It extends a pretrained CLIP ViT encoder with cross-attention modules that preserve strong visual features while effectively integrating tactile input. Please refer to **A2** in our response to Reviewer 5R2Q for further experimental details.
> > >
> > > While our core contributions lie in the hardware/system design and the in-the-wild dataset, we believe our model/pre-training design is also valuable to the community interested in representation learning from multi-modal, in-the-wild data involving complex physical interactions.
> > >
> > >
> > >
> > > >**Q2**: While the difference between this work and Vitamin is explained, the main difference lies in the sensor, which is not original and was proposed in previous works.
> > >
> > > We would like to clarify that ViTaMIn was uploaded to arXiv on April 8, 2025. According to NeurIPS’ official policy, papers appearing online after March 1, 2025, are considered contemporaneous and therefore cannot be grounds for rejection based on similarity. While both works share the high-level goal of enabling visuo-tactile learning for contact-rich manipulation using portable systems, we have already detailed several key differences in our response to Q1.
> > >
> > > Regarding the novelty of the sensor:  we made substantial improvements upon existing work. Our design is **novel** in that it uses a flexible PCB-based architecture, rather than relying on manually assembled sensors, significantly improving fabrication consistency and scalability. Furthermore, we are the first to demonstrate cross-pair transferability at scale: policies trained on one sensor pair can be seamlessly deployed on a different pair without retraining or adaptation. We provide quantitative results validating this consistency in **A9**, demonstrating a level of robustness and generalization not shown in prior work.
> > >
> > >
> > > >**Q3**:  The effectiveness of the in-the-wild data is not significant according to the experiment results.
> > >
> > >
> > > We appreciate the reviewer’s feedback and would like to clarify the impact of the in-the-wild data based on our results. Our experimental results show that **in-the-wild data has a clear and significant impact** on downstream task performance:
> > >
> > > - **Table 1** in the main paper reports that pretraining on our in-the-wild dataset improves task success rates by **25–60%**.
> > >
> > > - **Figure 6** further demonstrates that, in **low-data** and **low-epoch** regimes, **pretrained policies consistently outperform non-pretrained ones**.
> > >
> > > We would appreciate it if the reviewer could point out which specific aspects of these results they find unconvincing, as this would help us better understand and address the concern.
> > >
> > > Additionally, we believe the dataset itself provides unique value to the community due to its **scale**, **diversity**, and **unstructured real-world settings** that involve physical interactions, which remain underexplored in prior visuo-tactile learning work.

---

> > > > ### Comment · Reviewer_meGV · 2025-08-07
> > > >
> > > > Thank you for your feedback. My concerns have been clarified. One remaining concern is the comparison with the baseline: It could be more fair if the same backbone is used.

---

> ### Author Response · Authors · 2025-08-07
> **Official Comment by Authors**
>
> >**Q**: It could be more fair if the same backbone is used.
>
> **A**: Thank you for the continued feedback. We acknowledge that baseline methods from a few years ago may not use the most cutting-edge backbones. To ensure a fair comparison, we have tried to strengthen the baselines by replacing their original components with improved ones where appropriate while respecting their core designs. For example, we (1) matched M2VTP’s Transformer encoder size to that of our method, (2) adapted ViTAL’s multi-camera contrastive framework for our single-camera setup, and (3) aligned token sizes across all methods. Despite the efforts to strengthen the baselines, our model still outperforms them.
>
> Compared to M2VTP, which concatenates visual and tactile features and processes them jointly through self-attention, our approach separates the modalities and applies cross-modal attention. Vision and touch operate on different spatial and semantic scales. Empirically, **we observe that using two encoder branches with cross-attention leads to both better performance and more interpretable representations.**
>
> Compared to ViTAL, which uses contrastive learning between vision and touch, we found this approach to be unstable in our setting, even when using a strong vision encoder like CLIP. Empirically, we found **contrastive methods appear better suited for multi-camera setups with abundant observations, but tend to be unstable in our single-camera, in-the-wild setting.**
>
> We believe our model design **offers valuable practical insights for multi-modal representation learning from in-the-wild data involving complex physical interactions.** Our full model architecture and implementation details are detailed in Section 4 of the main paper and Section 4.2 of the Supplementary Material. We will fully open-source our code to support future research and community adoption.

---

### Official Review · Reviewer_MhLa · 2025-07-02

**Clarity:** 3
**Significance:** 2
**Originality:** 2
**Rating:** 4
**Confidence:** 4

**Summary:**

This paper presents a handheld (UMI-style) gripper system equipped with tactile sensing for collecting visuotactile demonstrations from human operators. The authors also propose a representation learning framework for learning visuotactile representations. The system is evaluated on four contact-rich tasks. While the approach is timely and results are promising, the paper leaves several key concerns unaddressed, and the overall novelty is limited.

**Questions:**

I would consider increasing the score if these concerns are clearly addressed:
* How do the authors justify collecting tactile signals through humans that receive no tactile feedback? In that case, how can the demonstrations reflect touch-driven behaviors?
* Is the encoder fine-tuned during imitation learning, or kept frozen?
* Are the evaluation tasks included in the pretraining data? Can you report performance when these tasks are held out?
* Are the hardware files and fabrication details of the handheld gripper going to be released?
* What limits the tactile sampling rate to 23Hz, and could it be increased?
* Would a stronger vision-only baseline with data augmentations reduce the observed performance gap?
* How reusable is the collected tactile data across sensors, especially considering variability in fabrication?

**Ethical Concerns:**

["NO or VERY MINOR ethics concerns only"]

**Final Justification:**

In general, I believe that limited novelty is the main remaining weakness of this work. But I think the paper has considerably improved during rebuttal and the work is overall well executed, so I will raise my score to a borderline accept.

**Limitations:**

Just about a line is dedicated to one single limitation of the approach. Please consider including some of the above weaknesses to provide readers with a clearer perspective on this work.

**Quality:**

3

**Strengths And Weaknesses:**

**Strengths:**

* Timely effort to enable large-scale visuotactile data collection using a handheld gripper interface.
* The system is demonstrated on contact-rich manipulation tasks, showcasing the importance of tactile sensing for representation learning.
* The authors will release the collected dataset, which should benefit the robotics community.

**Weaknesses:**

* A key issue with handheld grippers is the distribution mismatch between human-operated data collection and downstream robot deployment, which is not discussed or analyzed.
* The human teleoperator does not seem to receive tactile feedback. This raises concerns about the ability of the demonstrations to reflect touch-driven behavior, especially for tasks claimed to depend on tactile input (e.g., fluid transfer and whiteboard erasing). To further clarify, I agree that representations can still be stronger because of tactile sensing, but I am questioning the presence of touch-reactive behaviors.
* While the system is well-executed, the overall novelty is limited. The introduction lists 3 contributions. While equipping UMI with tactile sensing (1) is useful, contribution (2) is closely based on [22], and contribution (3) is a fairly standard application of masked autoencoding to visuotactile fusion [1],[2],[3], despite being claimed as novel.
* The encoder appears to be pretrained on data including the same tasks used for downstream evaluation. This makes it difficult to isolate the impact of pretraining. Holding out evaluation tasks during pretraining would provide a better perspective on generalization.

[1] VTAO-BiManip: Masked Visual-Tactile-Action Pre-training with Object Understanding for Bimanual Dexterous Manipulation. Sun et al., 2024.

[2] The Power of the Senses: Generalizable Manipulation from Vision and Touch through Masked Multimodal Learning. Sferrazza et al., 2023.

[3] Sparsh: Self-supervised touch representations for vision-based tactile sensing. Higuera et al., 2024.

---

> ### Author Rebuttal · Authors · 2025-07-31
>
> >**Q1**: A key issue with handheld grippers is the distribution mismatch between human-operated data collection and downstream robot deployment, which is not discussed or analyzed.
>
> **A1**: We took several steps to ensure consistency between the handheld gripper used for data collection and the robot platform used for deployment.
>
> (i) Camera Observation Alignment: A custom camera mount was built on the robot to match the height and orientation of the camera as it appears on the handheld gripper. During rollouts, we verified that the handheld and robot cameras captured nearly identical views.
>
> (ii) Gripper Width Calibration: We aligned the min/max widths of the handheld and XARM grippers, using linear interpolation to map between them. Calibration was verified both qualitatively (by finger spacing) and quantitatively (by replaying demonstrations).
>
> >**Q2**: The human teleoperator does not seem to receive tactile feedback. This raises concerns about the ability of the demonstrations to reflect touch-driven behavior, especially for tasks claimed to depend on tactile input (e.g., fluid transfer and whiteboard erasing). To further clarify, I agree that representations can still be stronger because of tactile sensing, but I am questioning the presence of touch-reactive behaviors.
>
> **A2**: The handheld gripper includes a built-in spring that provides force-based haptic feedback, allowing the human operator to feel resistance during grasping. This enables the operator to gauge how much force is being applied and whether the object is securely held.
> During downstream task data collection in the lab, we also provide real-time visualization of tactile signals on a screen, which the operator actively uses to guide interaction. In the fluid transfer task, where the pipette tip is occluded by water and not visible, the operator uses the tactile feedback to ensure that sufficient force is applied to extract the liquid successfully. Similarly, in the whiteboard erasing task, the operator both feels the contact through the spring mechanism and monitors the tactile signals on the screen to confirm that adequate pressure is being applied.
> Moreover, in the tube reorientation task, we conduct human-intervention experiments during policy execution. When the tube pose is intervened, the policy reorient the tube again. This demonstrates that the learned policy is responsive to tactile input.
>
> >**Q3**: While the system is well-executed, the overall novelty is limited. The introduction lists 3 contributions. While equipping UMI with tactile sensing (1) is useful, contribution (2) is closely based on [22], and contribution (3) is a fairly standard application of masked autoencoding to visuotactile fusion [1],[2],[3], despite being claimed as novel.
>
> **A3**:
>
> a) While our handheld device is inspired by [22], our work extends far beyond simply adding tactile sensors. We deliberately chose a piezoresistive tactile sensor for large-scale in-the-wild data collection due to its unique advantages: (i) Lightweight. (ii) Easy integration with soft grippers. (iii) We also hope to highlight another key advantage that may not be obvious at first glance on the **sensor consistency**. We designed and fabricated custom flexible PCBs to ensure sensor consistency, enabling cross-sensor transfer. This means we can collect data with one pair of sensors and successfully rollout policies with another sensor pair. Our design supports scaling to multiple data collection devices with different sensor pairs.
>
> b) While our masked autoencoding approach builds on a standard framework, our architecture is carefully designed to meet the unique challenges of a single-camera, handheld gripper setup. This targeted design enables it to outperform baselines like VITaL and MVT (see Q2 from Reviewer 5R2Q). We believe our approach offers meaningful insights for advancing visuo-tactile representation learning, particularly in the context of large-scale, in-the-wild datasets.
>
> c) Another core contribution of our work is large-scale visuo-tactile data collection in the wild, covering 43 diverse manipulation tasks. The dataset contributes unique value to the community in considering both vision and touch, and also physical interactions in the wild (not just touch as in paper [1]). We believe this scale and diversity largely advance the study of visuo-tactile correlation.
>
> [1] Tactile-Augmented Radiance Fields
>
>
> >**Q4**: The encoder appears to be pretrained on data including the same tasks used for downstream evaluation. This makes it difficult to isolate the impact of pretraining. Holding out evaluation tasks during pretraining would provide a better perspective on generalization.
>
> **A4**: We removed all test tube collection tasks from the pretraining dataset and did additional experiments. This results in a total of ~2500 in-the-wild videos for pretraining (previously ~2700). We pretrained the encoder on this reduced dataset, then finetuned it on the test tube collection task. Finally, we evaluated policy performance.
> i) Qualitative Observations: The attention heatmap on both the training and validation set images remains primarily focused on the gripper region.
> ii) Quantitative Result: We conducted policy rollouts for the test tube collection task for both policies. We observed no behavioral inconsistencies between the two policies:
> | Task                                                       | Grasp | Reorient | Insert | Whole Task |
> |------------------------------------------------------------|-------|----------|--------|-------------|
> | Policy w/ pretraining (ours)                               | 1.0   | 1.0      | 0.85   | 0.85        |
> | Policy w/ pretraining (without tube in pretrain data)      | 1.0   | 1.0      | 0.80   | 0.80        |
>
> >**Q5**: How do the authors justify collecting tactile signals through humans that receive no tactile feedback? In that case, how can the demonstrations reflect touch-driven behaviors?
> **A5**: Please see our response to **Q2**
>
> >**Q6**: Is the encoder fine-tuned during imitation learning, or kept frozen?
>
> **A6**: The encoder is fine-tuned during imitation learning. We believe fine-tuning is important because, while the pretrained encoder captures general visuo-tactile relationships, individual tasks often require additional task-specific features from both modalities. To preserve the benefits of pretraining while allowing adaptation, we fine-tune the encoder using a smaller learning rate (3e-5) compared to the diffusion module (3e-4).
>
> >**Q7**: Are the evaluation tasks included in the pretraining data? Can you report performance when these tasks are held out?
>
> **A7**: Please see our response to **Q4**
>
> >**Q8**: Are the hardware files and fabrication details of the handheld gripper going to be released?
>
> **A8**: Yes, we will release the complete hardware setup, including all fabrication details and files, to enable full reproducibility of our handheld gripper system.
>
> >**Q9**: What limits the tactile sampling rate to 23Hz, and could it be increased?
>
> **A9**: The tactile sampling rate is currently limited to 23 Hz due to constraints in our PCB design and firmware. It can be increased through several possible improvements:
>
> (i) **Upgrading MCU** (current Arduino Nano) with a higher-performance microcontroller would directly enhance sensor reading speed.
>
> (ii) **Data compression** for serial communication in firmware using. For instance, instead of individually transmitting long sequences of zero-valued readings, we can encode consecutive zeros compactly (e.g., transmitting “17 zeros” rather than seventeen individual zeros).
>
> (iii) Employing **subsampling and reconstruction** methods by randomly subsampling sensor measurements and reconstructing full tactile frames.
>
> >**Q10**: Would a stronger vision-only baseline with data augmentations reduce the observed performance gap?
>
> **A10**: Our vision-only baseline follows the exact implementation from Chi et al. in the UMI paper. Specifically, we finetune from a pretrained CLIP vision encoder and apply standard image augmentations, including color jittering and spatial cropping, consistent with prior work and shown to be effective.
>
> All four policies in our comparison use the same vision augmentation setup to ensure fairness. While additional augmentations might offer marginal improvements, we believe they are unlikely to close the performance gap. Most failure cases in the vision-only baseline occur in scenarios where tactile feedback is essential—for example, determining whether a test tube is upright or tilted—especially given its transparency—or extracting the correct amount of fluid with a pipette, which requires precise force control. These challenges are inherently difficult to solve with vision alone.
>
> >**Q11**: How reusable is the collected tactile data across sensors, especially considering variability in fabrication?
>
> **A11**: During data collection and evaluation, we used four tactile sensors across two different grippers and robot arms. All sensors were fabricated in the same batch using an identical production process, and we observed consistent performance across all units.
>
> In addition to this consistency, our sensor design focuses on capturing structural tactile patterns—such as contact distribution and pressure regions—rather than relying on fine-grained signal details. This makes our sensor inherently robust to minor variations between sensor units, which supports generalization across different hardware setups.
>
> >**Q12 Limitations**: Including some of the above weaknesses to provide readers with a clearer perspective on this work.
>
> **A12**:
>
> - Careful manual calibration is still required to align the physical setups of the handheld and robot grippers.
>
> - We still need to stream tactile data to a host Ubuntu machine using USB. Enabling wireless communication  would improve portability.
>
> - Tactile reading is limited to 23 hz for current PCB Design.

---

> > ### Comment · Reviewer_MhLa · 2025-08-04
> >
> > Thank you for responding to my concerns. I have some follow-up questions and comments:
> >
> > Q1: I understand these calibration steps are helpful. I was referring more to the fact that human and robot arm morphologies are different. E.g., how do you ensure that the trajectory followed by the human during data collection can actually be followed by the robot without colliding with the environment? Does data collection require filtering of the trajectories due to this issue?
> >
> > Q3: a) this is an interesting point, and it would have actually been more convincing to see cross-sensor generalization experiments; b) as stated earlier, all components are quite standard and the contribution is not fully convincing. That said, the new experiments are helpful.
> >
> > Q4: Performance seems to decrease, so it would be interesting to see more 'held-out' results.
> >
> > Q2, Q6, Q8, Q9, Q10: It would be helpful to have these clarifications in the next version of the paper.

---

> > > ### Author Response · Authors · 2025-08-08
> > > **Official Comment by Authors**
> > >
> > > Thank you again for your constructive reviews and suggestions. Since we are one day away from the Author-Reviewer discussion deadline, we wanted to follow up to see if our latest batch of responses has addressed your concerns or if we can provide any additional clarification. Thank you.

---

> ### Author Response · Authors · 2025-08-05
> **Official Comment by Authors**
>
> >**Q1**: I understand these calibration steps are helpful. I was referring more to the fact that human and robot arm morphologies are different. E.g., how do you ensure that the trajectory followed by the human during data collection can actually be followed by the robot without colliding with the environment? Does data collection require filtering of the trajectories due to this issue?
>
> **A1**: This has been a well-known issue in handheld gripper setups like UMI, that human and robot arm morphologies differ and that not all human demonstrations can be replicated by the robot (e.g., rapid wrist movements).
>
> We did not apply explicit filtering to the collected data. Instead, in the early stage of the project, we went through several iterations of collecting trajectories and replaying them on the robot to ensure they could be executed accurately and reliably. These iterations were similar to human teleoperation, though not as real-time. This replaying step was not redundant; it also allowed us to verify whether the SLAM system produced correct and reasonable trajectories. Through this process, human demonstrators developed an intuitive understanding of the types of trajectories the robot could successfully execute, which naturally informed how demonstrations were performed in later stages.
>
>
> >**Q3**: a) this is an interesting point, and it would have actually been more convincing to see cross-sensor generalization experiments; b) as stated earlier, all components are quite standard and the contribution is not fully convincing. That said, the new experiments are helpful.
>
> **A3**:
> The sensor pair in the handheld gripper differs from the one used in the robot gripper, meaning that we have already demonstrated cross-sensor generalization. To further validate the consistency of our tactile sensors, we conducted additional experiments today by replacing two of the robot’s sensors with newly fabricated ones that had not been used during the project. We then evaluated the test tube collection task (trained on data from a different sensor pair) using these new sensors.
>
>
> | Task                               	                | Grasp  | Reorient | Insert | Whole Task |
> |-------------------------------------------------------|--------|----------|--------|-------------|
> | Policy w/ pretraining (ours)                          | 1.00 | 1.00   | 0.85 | 0.85      |
> | Policy w/ pretraining (new sensors)                   | 1.00 | 1.00   | 0.85 | 0.85      |
>
> The results suggest that our sensors are consistent enough to support effective downstream task performance across different sensor pairs.
>
> To quantitatively evaluate sensor consistency, we refer to the experiment described in our response to Reviewer meGV (A9) in the following, where we show that the statistical distributions of sensor signals from human demonstrations and robot rollouts, each using different sets of sensors, are closely aligned across all evaluated metrics. This demonstrates consistency both across sensor pairs and between human and robot behaviors.
>
> Quantitative Results:
>
> We processed 60 Hz tactile data from training and recorded 10 Hz tactile signals across 20 successful rollouts. We then compared the statistical distributions of training and rollout signals, confirming their alignment.
>
> ### Per-pixel Distribution Statistics
>
> | Split | n       	| mean   | std	| min	| max	| skew   | kurt   |
> |-------|-------------|--------|--------|--------|--------|--------|--------|
> | Train (Human Demonstrations) | 286,114,560 | 0.0461 | 0.1699 | 0.0000 | 1.0000 | 3.9312 | 14.8065 |
> | Eval (Robot Rollouts) | 1,324,800   | 0.0414 | 0.1523 | 0.0000 | 1.0000 | 4.0988 | 16.8261 |
> —
>
> ### Per-episode Distribution Statistics
>
> | Split          	| mean   | std of means | median | 95%ile  |
> |--------------------|--------|---------------|--------|---------|
> | Train episode-means (Human Demonstrations) | 0.0461 | 0.0035    	| 0.0465 | 0.0512  |
> | Eval episode-means (Robot Rollouts) | 0.0414 | 0.0036    	| 0.0409 | 0.0477  |
>
> Results show similar tactile signal levels between train (mean = 0.046, std = 0.170) and eval (mean = 0.041, std = 0.152), with tight episode-level variability (train std = 0.0035; eval = 0.0036). Both distributions exhibit strong right skew and high kurtosis, indicating the quality of our tactile signals that are consistent across sensors and between human demonstrations and robot behaviors.
>
> >**Q4**: Performance seems to decrease, so it would be interesting to see more 'held-out' results.
>
> **A4**: We are currently training on another task using held-out data and will share the results tomorrow.
>
>
> >**Q2, Q6, Q8, Q9, Q10**: It would be helpful to have these clarifications in the next version of the paper.
>
> **A**: Thank you for the constructive suggestions. We will incorporate these clarifications into the main paper in the final version.

---

> > ### Author Response · Authors · 2025-08-06
> > **Response to Q4: Additional Task Results**
> >
> > >**Q4**: Performance seems to decrease, so it would be interesting to see more 'held-out' results.
> >
> > **A4**: We repeated the held-out experiment on the whiteboard erasing task by removing all corresponding demonstrations from the pretraining dataset. After this removal, the pretraining dataset contained approximately 2500 videos (previously ~2700 videos).
> >
> > i) Qualitative Observations: Similarly, the attention heatmap on both the training and validation set images remains primarily focused on the gripper region.
> >
> > ii) Quantitative Result: We conducted policy rollouts for the whiteboard erasing task for both policies. Similarly, we observed no behavioral inconsistencies between the two policies:
> >
> > | **Task**                                               | **1st Erase** | **2nd Erase** | **Whole Task** |
> > |--------------------------------------------------------|-----------------|------------------|----------------|
> > | Policy w/ pretraining (ours)                           | 0.70            | 0.75             | 0.70           |
> > | Policy w/ pretraining (without erase in pretrain data) | 0.75            | 0.75             | 0.75           |

---

> ### Comment · Reviewer_MhLa · 2025-08-08
>
> Thank you again for engaging in the rebuttal and considering my comments.
>
> Q1: While this might a known issue for those working directly with handheld grippers, it is still important to point it out, and indeed the process you describe is quite interesting. Readers want to learn as much as possible from papers, which is why I think it's relevant to not only list results, but also limitations, insights, practices learned during the process.
>
> Q4: Thank you for providing this additional ablation. This is more convincing, and I would invite the authors to still add a full held-out analysis in the final version of this paper.
>
> In general, I believe that limited novelty is the main remaining weakness of this work. But I think the paper has considerably improved during rebuttal and the work is overall well executed, so I will raise my score to a borderline accept.

---

### Official Review · Reviewer_5R2Q · 2025-07-02

**Clarity:** 3
**Significance:** 3
**Originality:** 3
**Rating:** 5
**Confidence:** 5

**Summary:**

This paper presents a portable tactile gripper designed for in-the-wild data collection and manipulation. The authors develop the tactile sensors compatible with parallel grippers and propose a representation learning framework that aligns tactile and visual modalities through masked reconstruction. The system is pretrained on a large dataset of over 500K visuo-tactile pairs collected across 43 diverse tasks and later fine-tuned on four downstream manipulation tasks. The experiments demonstrate clear benefits of using tactile information and using the visuo-tactile pretraining in improving policy robustness and success rates, particularly in contact-intensive and visually ambiguous settings.

**Questions:**

Can you quantify the contribution of the tactile sensor’s spatial resolution and time synchronization to downstream policy success?

How does your pretraining compare quantitatively against existing visual-tactile methods such as VITaL or MVT?

Do the vision encoders (in either pretrained or baseline conditions) use CLIP? If so, does this conflate the benefits of vision-language pretraining with visuo-tactile learning?

Can you provide a more detailed visualization or summary of the 43 tasks used for pretraining?

**Ethical Concerns:**

["NO or VERY MINOR ethics concerns only"]

**Final Justification:**

The authors provide a comprehensive rebuttal and I believe my concerns have been adequately addressed, so I will maintain my acceptance rating.

**Limitations:**

While the system is capable of collecting tactile data in unstructured environments, the method is currently limited to hand-held grippers and parallel jaw designs. It may not generalize to whole-body or multi-fingered manipulation without significant redesign. Additionally, policy learning still requires domain-specific downstream fine-tuning.

**Paper Formatting Concerns:**

No Concerns.

**Quality:**

3

**Strengths And Weaknesses:**

Strength:

Hardware Contribution. The design and deployment of a compact, vision-aligned tactile sensor for use on handheld grippers is a great engineering achievement. It enables scalable collection of visuo-tactile data in the wild and lowers the barrier for tactile policy learning.

Data Collection at Scale. The authors curate a large and diverse dataset spanning 43 tasks across real-world environments, which is beneficial to the community.

Cross-Modal Pretraining. The paper makes a compelling case for joint visuo-tactile representation learning via masked reconstruction. The ablations convincingly show the advantages of pretraining for sample efficiency, attention localization, and final task performance.

Detailed Experiments. The evaluation across four long-horizon manipulation tasks is extensive. The attention maps, loss curves, and failure analyses provide valuable insights into how and when tactile feedback contributes to robustness.

Weakness:

Sensor Evaluation. While the tactile sensor design is innovative, the paper lacks quantitative evaluation of the sensor’s improvements. More specifically, how does the improved tactile resolution and time synchronization affect manipulation accuracy?

Baselines: Although the paper presents comprehensive ablation experiments. The baselines for visuo-tactile representation learning are relatively weak. Comparisons against recent visual-tactile methods (e.g., Masked Visual-Tactile Pretraining [1], VITaL [2]) would strengthen claims of superiority.

[1] Masked Visual-Tactile Pre-training for Robot Manipulation.
[2] VITaL Pretraining: Visuo-Tactile Pretraining for Tactile and Non-Tactile Manipulation Policies

Visualization of Tasks: While the paper mentions 43 tasks for pretraining, a more detailed breakdown or visualization of task categories and environments (beyond Figure 5) would help clarify dataset diversity.

CLIP Usage Ambiguity. It is unclear whether the “no pretraining” baseline still benefits from a pretrained CLIP vision encoder. If so, this should be clarified to disentangle the benefit of cross-modal pretraining from that of strong visual initialization.

---

> ### Author Rebuttal · Authors · 2025-07-31
>
> >**Q1**: Sensor Evaluation. While the tactile sensor design is innovative, the paper lacks quantitative evaluation of the sensor’s improvements. More specifically, how does the improved tactile resolution and time synchronization affect manipulation accuracy?
>
> **A1**: We appreciate the reviewer’s positive comments regarding our tactile sensor design.
>
> a) Quantitative Results for sensor resolution:
>
> To assess the impact of tactile spatial resolution on downstream manipulation performance, we perform 20 rollouts using our policy with pretraining, each with reduced-resolution tactile input 6×16 for each finger (original 12×32). Initial conditions are kept consistent, as described in the main paper.
>
> | Task                                     | Grasp | Reorient | Insert | Whole Task |
> |------------------------------------------|-------|----------|--------|-------------|
> | Policy w/ pretraining (ours)             | 1.0   | 1.0      | 0.85   | 0.85        |
> | Policy w/ pretraining (low-res tactile)  | 1.0   | 1.0      | 0.75   | 0.75        |
>
> Qualitative Observations: With lower tactile resolution, we observed a higher likelihood of misalignment during the insertion phase. This is likely due to the coarser spatial granularity: the downsampled tactile signal fails to precisely capture the contact location with the test tube, making it more difficult for the policy to localize contact and perform accurate insertions with tight clearance. We also noticed increased hesitation during reorientation, as the reduced resolution introduces uncertainty in state estimation—similar to behaviors observed in vision-only policies.
>
> b) Synchronization:
>
> We refer to the UMI paper (Chi et al.) which conducted experiments comparing latency matching (synchronized observation) and no latency matching (non-synchronized observations). Their results showed that aligning sensor latency led to approximately a 30% improvement in performance, highlighting the importance of accurate time synchronization in multi-modal policy learning.
>
> >**Q2**: Baselines: Although the paper presents comprehensive ablation experiments. The baselines for visuo-tactile representation learning are relatively weak. Comparisons against recent visual-tactile methods (e.g., Masked Visual-Tactile Pretraining [1], VITaL [2]) would strengthen claims of superiority.
>
> **A2**: We appreciate the suggestion and have conducted thorough comparisons by faithfully reimplementing and adapting M2VTP and VITaL to our pipeline for a fair evaluation. Our method achieves higher success rate and leads to faster convergence and consistently reaches a lower final loss compared to baselines. The following tables show the downstream tasks loss and success rate for ours and two baselines for test tube collection task:
>
> **Downstream Tasks Loss**
>
> | Epoch    | MAE (Ours) | M2VTP     | VITaL     |
> |----------|-------------|-----------|-----------|
> | Epoch 10 | **0.013352** | 0.014486  | 0.014425  |
> | Epoch 20 | **0.012058** | 0.013864  | 0.013885  |
> | Epoch 30 | **0.011490** | 0.013360  | 0.013529  |
> | Epoch 40 | **0.011196** | 0.012967  | 0.013142  |
> | Epoch 50 | **0.010870** | 0.012537  | 0.012802  |
> | Epoch 60 | **0.010569** | 0.012164  | 0.012492  |
>
> **Downstream Tasks Success Rate**
>
> | Task                                     | Grasp | Reorient | Insert | Whole Task |
> |------------------------------------------|-------|----------|--------|-------------|
> | MAE (ours)                                | 1.0   | 1.0      | 0.85   | 0.85        |
> | M2VTP                                      | 1.0   | 0.9      |  0.1   |   0.1      |
> | VITaL                                         | 0.1  | 0.0      | 0.0   | 0.0        |
>
> **Detailed Analysis**: Our analysis shows that these methods are tailored to fundamentally different use cases and are not well-suited to the demands of single-camera, in-the-wild manipulation, which is the focus of our work.
>
> a) M2VTP: M2VTP pretrains a masked autoencoder with a shared transformer that processes 224×224 RGB images and tactile signals, embedded into 768D tokens. After partial masking and reconstruction of both modalities, the encoder is used as a feature extractor for reinforcement learning-based dexterous control.
>
> - i) Qualitative Observations: M2VTP achieves strong tactile reconstruction, but even after extensive training and convergence, its image reconstructions remain coarse, particularly in the gripper's central region and corresponding object. This may be due to the diversity of in-the-wild data and the method’s reliance on large image masking.
>
> - ii) Downstream task performance: M2VTP is designed for simulated tabletop tasks with a dexterous hand and uses a transformer-based vision encoder trained from scratch, which limits its capacity and generalization to in-the-wild settings. Its use of heavy image masking and weak focus on visual reconstruction, likely beneficial in settings that prioritize tactile contact, further reduces sensitivity to fine visual details. In our setting, while M2VTP performs well on grasping and reorientation, it completely fails on insertion tasks for transparent objects, which demand precise visual feedback.
>
> b) VITaL: VITaL uses two ResNet-18 encoders for vision and tactile inputs to produce 512-dimensional features. These are projected into a shared space via MLPs and trained using a temporally-aware contrastive loss. Post-training, the encoder is used in imitation learning frameworks.
>
> - i)  Qualitative Observations: VITaL requires roughly 3 times longer training on the same dataset and shows less stable pretraining performance, with noticeably noisier validation curves and larger standard deviations across different data batches.
>
> - ii) Downstream task performance: VITaL is a contrastive learning method tailored for multi-camera setups, where contrastive loss is computed separately for each camera view. It uses ResNet-18 encoders for both vision and tactile inputs, trained from scratch. The vision encoder is 7× smaller than our CLIP ViT-B/16, severely limiting visual capacity, especially under single-camera, in-the-wild conditions. Moreover, using a full ResNet to encode our 12*32 tactile is unnecessarily complex and inefficient. Despite faithfully reimplementing the full pipeline, VITaL was unsuccessful in rollout, as its trajectories were consistently noisy and failed to complete most task steps. The method is fundamentally mismatched to the demands of large-scale, single-camera, real-world manipulation.
>
> >**Q3**: Visualization of Tasks: While the paper mentions 43 tasks for pretraining, a more detailed breakdown or visualization of task categories and environments (beyond Figure 5) would help clarify dataset diversity.
>
> **A3**: We appreciate the reviewer’s interest in the task diversity of our dataset. We have visualized it, but we may not provide figure during rebuttal due to policy. Below is a concise breakdown of the 43 tasks grouped by environment:
>
> **Indoor (Main Tasks)**:
>
> Test Tube Collection, Fluid Transfer, Whiteboard Erasing, Pencil Insertion
>
> **Indoor (Other Tasks)**:
>
> Write on Whiteboard, Pick & Place Circular Objects, Pick & Place Cup, Toss SD Card, Peg Insertion, Hex Key Insertion
>
> **In-the-Wild Tasks**:
>
> - **Chipotle**: Mix Bowl
> - **Starbucks**: Place Plate, Insert Straw
> - **Makerspace**: Reorient Marker, Place Plastic Thread into Box, Place Glass into Glass Box, Put Tool into Solder Holder, Turn Soldering Machine Button, Plug USB, Organize Paper Rolls, Place Hex Key, Shut LEGO Box Lid, Assemble LEGO Brick, Insert Metal Socket into Set
> - **Sink**: Hang Cleaning Spray, Reorient Paint Brush
> - **Kitchen**: Open Microwave, Open Cabinet, Plug Cord
> - **Library**: Sort Books
> - **Hardware Store**: Hang Product Pads, Organize Canvas Items, Place Foam Brush, Place Wooden Box, Place Glass Cup, Place Vaseline, Move Books
> - **Campus – Table**: Pour Water, Place Saucer Cup
> - **Campus – Staircase**: Write Letter “R”
> - **Grassland**: Throw Squeeze Ball
> - **Student Lounge**: Hang Public Phone, Grasp Food
>
> >**Q4**: CLIP Usage Ambiguity. It is unclear whether the “no pretraining” baseline still benefits from a pretrained CLIP vision encoder. If so, this should be clarified to disentangle the benefit of cross-modal pretraining from that of strong visual initialization.
>
> **A4**: All vision encoders in our experiments, both in the baseline and visuo-tactile pretraining conditions, use a CLIP backbone. This follows established practice (e.g., UMI) and provides a strong, consistent visual representation across all policies.
>
> We would like to clarify that this setup does not conflate the benefits of vision-language pretraining (from CLIP) with those of visuo-tactile learning. In our work, “pretraining” refers specifically to visuo-tactile pretraining, where the model aligns visual and tactile inputs through joint representation learning. Both baseline and pretrained models use the same CLIP-initialized vision encoder; the only difference is whether this encoder is further trained with tactile context. This design isolates the contribution of visuo-tactile pretraining: since all models use CLIP, performance gains reflect the value of cross-modal alignment, not the vision backbone alone.
>
> >**Q5**: Can you quantify the contribution of the tactile sensor’s spatial resolution and time synchronization to downstream policy success?
>
> **A5**: Please see our response to **Q1**.
>
> >**Q6**: How does your pretraining compare quantitatively against existing visual-tactile methods such as VITaL or MVT?
>
> **A6**: Please see our response to **Q2**.
>
> >**Q7**: Do the vision encoders (in either pretrained or baseline conditions) use CLIP? If so, does this conflate the benefits of vision-language pretraining with visuo-tactile learning?
>
> **A7**:  Please see our response to **Q4**.
>
> >**Q8**: Can you provide a more detailed visualization or summary of the 43 tasks used for pretraining?
>
>
> **A8**: Please see our response to **Q3**

---

> > ### Author Response · Authors · 2025-08-05
> > **Official Comment by Authors**
> >
> > Thank you for the constructive reviews. We'd like to follow up and check if our rebuttal has addressed your concerns. Please let us know if there are any remaining questions or if we can provide additional clarification. We're happy to elaborate further if helpful.

---

### Official Review · Reviewer_Tduh · 2025-07-06

**Clarity:** 3
**Significance:** 3
**Originality:** 4
**Rating:** 5
**Confidence:** 3

**Summary:**

This paper presents a complete, end-to-end solution for the challenging problem of learning fine-grained robotic manipulation in unstructured, "in-the-wild" environments. The authors begin by designing and implementing a portable, handheld gripper integrated with piezoresistive tactile sensors, which enables the synchronized collection of visual and high-resolution tactile data. Building upon this hardware, the paper proposes a novel cross-modal representation learning framework, centered around a masked tactile reconstruction self-supervised task, designed to effectively fuse global context from vision with local contact details from touch. Finally, the learned multimodal representations are integrated into a Diffusion Policy and validated on a series of challenging tasks where subtle force feedback and control are needed, such as "Transparent Tube Collection" and "Fluid Transfer." The experimental results clearly demonstrate the significant superiority of this method compared to vision-only baselines and other simpler fusion approaches.

**Questions:**

see the weakness

**Ethical Concerns:**

["NO or VERY MINOR ethics concerns only"]

**Limitations:**

The authors provide a thorough analysis of the potential limitations of their method.

**Paper Formatting Concerns:**

I did not notice any major formatting issues.

**Quality:**

4

**Strengths And Weaknesses:**

Strengths

1. This work accurately identifies a key deficiency in current large-scale teleoperation systems, which is crucial for fine-grained manipulation. The proposed portable visuo-tactile gripper is a highly practical and well-engineered solution. Its lightweight design (approx. 962g), high-resolution flexible tactile sensors, and innovative hardware-free data synchronization scheme together constitute a powerful data collection tool that greatly facilitates community research on contact-rich manipulation tasks.

2. In contrast to common approaches like contrastive learning or simple feature concatenation, the authors employ a masked auto-encoding objective. This method cleverly forces the model to understand the intrinsic physical patterns of tactile signals and learn how to infer missing tactile information from visual context. As shown in Qualitative Results of Pre-Training(Figure 4), this approach not only reconstructs tactile images accurately but also learns more interpretable features—the vision encoder's attention consistently focuses on the contact regions between the gripper and the object, which is an elegant and effective contribution.

3. The experimental results in Table 1 are highly persuasive. The proposed method consistently outperforms all baselines across the board. It shows significant improvement not only over the Vision Only approach (achieving an 85% vs. 25% success rate in the tube collection task, for example) but also clearly surpasses a naive CNN Tactile Fusion method and its own architectural variant without pre-training. Furthermore, the ablation study in Figure 6 clearly demonstrates the immense value of pre-training, showing that the pre-trained model achieves far superior performance in low-data and low-epoch training settings.

4. The authors' commitment to open-sourcing their hardware design, code, and a large-scale visuo-tactile dataset across 43 task is a valuable contribution. This will significantly enhance reproducibility and drive future work in the field, an effort worthy of high praise.

Weaknesses

1. Piezoresistive sensors can experience performance drift under long-term mechanical stress and in varying environmental conditions, such as temperature and humidity. The paper could be strengthened by discussing the long-term durability of the sensors. This would make the hardware contribution more complete and practical.

2. The paper reports high success rates, but a more in-depth qualitative analysis of failure cases would be beneficial. For example, what is the root cause of the 10% of failures in the "Fluid Transfer" task in Table 1? Providing a visual analysis of failure cases (e.g., with corresponding tactile signals) would offer deeper insights.

---

> ### Author Rebuttal · Authors · 2025-07-31
>
> >**Q1**: Piezoresistive sensors can experience performance drift under long-term mechanical stress and in varying environmental conditions, such as temperature and humidity. The paper could be strengthened by discussing the long-term durability of the sensors. This would make the hardware contribution more complete and practical.
>
> **A1**: We appreciate the reviewer’s thoughtful question regarding long-term durability. We specifically designed our piezoresistive tactile sensors using flexible PCBs to enhance mechanical robustness and ensure consistency across sensor batches. In practice, we have distributed our sensors to external labs, where they have been used continuously for nearly a year without noticeable performance degradation or failure. Regarding environmental factors:
>
> (i) Humidity: We have not observed any performance issues related to humidity during extensive in-the-wild data collection across both indoor and outdoor environments.
>
> (ii) Temperature: Like most piezoresistive materials, the sensing layer can exhibit signal drift under elevated temperatures. While our sensors operate reliably under typical usage conditions, extremely high temperatures may impact performance. Many commercial piezoresistive tactile sensors are rated for operation within the range of –40 °C to 60 °C, and we expect similar properties from our design, as it shares the same material foundation.
> >**Q2**: The paper reports high success rates, but a more in-depth qualitative analysis of failure cases would be beneficial. For example, what is the root cause of the 10% of failures in the "Fluid Transfer" task in Table 1? Providing a visual analysis of failure cases (e.g., with corresponding tactile signals) would offer deeper insights.
>
> **A2**: We list the common failure cases observed with the policy with pretraining:
>
> - Test Tube Collection: The policy misaligns the test tube with the rack hole, resulting in collision with the rack edge.
>
> - Fluid Transfer: During the expelling stage, the pipette occasionally fails to fully release the liquid, leaving some water inside.
>
> - Pencil Insertion: The robot reaches the vicinity of the insertion hole but fails to achieve precise alignment, which causes the insertion failure.
>
> - Whiteboard Erasing: The robot may rotate to an incorrect position, misaligning the eraser with the text and leading to incomplete erasure.
>
> We have already created visualizations that pair failure cases with corresponding tactile signals. While we cannot include external links due to rebuttal policy, we will incorporate representative visualizations in the revised versions.

---

> > ### Author Response · Authors · 2025-08-05
> > **Official Comment by Authors**
> >
> > Thank you for the constructive reviews. We'd like to follow up and check if our rebuttal has addressed your concerns. Please let us know if there are any remaining questions or if we can provide additional clarification. We're happy to elaborate further if helpful.

---

### Note · Authors · 2025-08-12

We thank all reviewers for their constructive and detailed feedback, which has improved the clarity and thoroughness of our work.
In summary, in response to the reviews, we have improved our paper as follows:

 (1) Conducted quantitative statistical comparisons of tactile signal distributions across different sensor pairs and evaluated downstream performance after sensor replacement. Both analyses show that **cross-sensor data are well aligned** and that **downstream performance is unaffected**, demonstrating **strong cross-pair transferability at scale.**

 (2) Added comparisons with recent multi-modal baselines (M2VTP, VITaL) using matched configurations, as suggested by Reviewer 5R2Q. Downstream results show that **our pretraining is significantly more effective for single-camera, in-the-wild setups than other baselines.**

 (3) Conducted in-the-wild pretraining experiments (tube collection, whiteboard erasing) with held-out evaluation tasks to validate the usefulness of in-the-wild data. Similar downstream performance confirms the **strong value of in-the-wild data for downstream task learning.**

 (4) Detailed dataset diversity across 43 in-the-wild tasks.

We note two remaining questions from Reviewer meGV, for which we provided clarifications but did not receive responses:

 (1) **Comparison with Contemporaneous Works.** ViTaMIn was uploaded to arXiv on Apr 8, 2025, and per NeurIPS policy, contemporaneous works (post–Mar 1, 2025) cannot be grounds for rejection based on similarity. While both papers address visuo-tactile learning with portable systems, we detailed substantial differences in hardware, dataset, and downstream tasks in Q1 to Reviewer meGV.

 (2) **Baselines with Same Backbones.** To ensure a fair comparison, we strengthened baselines by (i) matching M2VTP’s Transformer encoder size to ours, (ii) adapting ViTAL’s multi-camera contrastive framework to our single-camera setup, and (iii) aligning token sizes across all methods while preserving their core designs. Even with these adjustments, our method consistently outperforms the baselines.

We appreciate the reviewers’ engagement throughout the process and will incorporate all clarifications and additional experiments into our revised versions. **We are committed to fully open-sourcing all hardware, code, and datasets to maximize community benefit.**

---

### Decision · Program_Chairs · 2025-09-17

**Decision:**

Accept (poster)

**Comment:**

This paper presents a practical, portable visuo-tactile handheld gripper and a simple two-stage learning recipe (masked tactile reconstruction with cross-modal fusion to diffusion policy) that together deliver clear gains on contact-rich, partially occluded manipulation tasks collected “in the wild.” Reviewers were mixed on technical novelty—arguing the learning pieces are incremental and contemporaneous with related work—but were uniformly positive about the engineering value, the sizeable synchronized RGB-tactile dataset, and the convincing real-robot improvements over vision-only and naïve fusion baselines. The rebuttal strengthened the case by adding stronger baselines (e.g., VITaL, M2VTP) under matched settings, clarifying CLIP usage, and providing useful durability notes; while some risks remain (breadth limited to four tasks on one platform, modest gains on one task, and missing uncertainty estimates), these are addressable in camera-ready.

I encourage the authors to expand comparisons to concurrent methods, report confidence intervals for success rates, include failure-case visualizations with tactile traces, and summarize calibration/drift guidance.

Overall, the combination of accessible hardware, real-world data, and consistent empirical improvements makes this a valuable contribution for the community. Recommendation: Accept.